# Quantum speedup for nonreversible Markov chains

Baptiste Claudon [1,2,3] ✉, Jean-Philip Piquemal [1,3] ✉ &
Pierre Monmarché [2,3,4,5] ✉

Quantum algorithms can potentially solve a handful of problems more efficiently than their classical counterparts. In that context, it has been discussed that Markov chains problems could be solved significantly faster using quantum computing. Indeed, previous work suggests that quantum computers could accelerate sampling from the stationary distribution of reversible Markov chains. However, in practice, certain physical processes of interest are nonreversible in the probabilistic sense and reversible Markov chains can sometimes be replaced by more efficient nonreversible chains targeting the same stationary distribution. This study constructs Markov chain reversibilizations and develops quantum algorithmic techniques to accelerate nonreversible processes. Such an up-to-exponential quantum speedup goes beyond the predicted quadratic quantum acceleration for reversible chains and is likely to have a decisive impact on many applications ranging from statistics and machine learning to computational modeling in physics, chemistry, biology and finance.

Quantum algorithms recently gained attention due to their asymptotic advantage over the best known classical methods for certain problems[1,2]. Such speedups turned quantum computing into an active field of research. Monte Carlo simulations both offer a wide range of applications and promise to be accelerated by quantum algorithms[3]. The speedup is typically provided by the Quantum Amplitude Estimation (QAE) routine[4,5]. The routine requires access to samples from a target probability distribution $\pi$. When dealing with complex, high-dimensional distributions known only up to a normalization constant (as in statistical physics), Markov Chain Monte Carlo (MCMC) is among the most popular methods to provide such classical samples[6]. If not used solely, it is often part of more elaborate algorithms such as Sequential Monte Carlo[7] or Annealed Importance Sampling[8]. The quantum coherent state $|\pi\rangle$ associated with the distribution may be prepared using the Quantum Rejection Sampling (QRS) algorithm[9], but QRS requires bounds on the normalization constants for efficiency. The present introductory paragraph uses common definitions, which are recalled in the "Methods" section "Markov chains". In

MCMC, we start by designing an ergodic Markov chain that converges to the sought distribution $\pi$[10]. Performing sufficiently many steps of the chain yields an approximate sample from $\pi$. The main difficulty is that the mixing time, the minimum number of steps for the sample to be distributed according to $\pi$, may be large. It is well-known that, to each reversible chain, we can associate a quantum walk with quadratically larger spectral gap[11]. It is then possible to construct the reflection through the stationary distribution in the inverse spectral gap of this quantum walk (see, for example[12,13]). Since the mixing time is of the order of the inverse spectral gap of the chain, the reflection can readily be used to provide a sample from $\pi$ with quadratic speedup[14].

First and foremost, we are often interested in studying physical processes that are nonreversible in the probabilistic sense. Such processes may have a known stationary distribution, as for the underdamped Langevin dynamics, or not, as in the study of out-of-equilibrium systems in statistical physics[15,16]. In addition, reversible chains are known to show a diffusive behavior[17]. It is sometimes

¹Advanced Research Department, Qubit Pharmaceuticals, Paris, France. ²Sorbonne Université, LJLL, UMR 7198 CNRS, Paris, France. ³Sorbonne Université, LCT, UMR 7616 CNRS, Paris, France. ⁴Institut Universitaire de France, Paris, France. ⁵Present address: LAMA, Université Gustave Eiffel, Marne-la-Vallée, France. ✉e-mail: baptiste.claudon@qubit-pharmaceuticals.com; jean-philip.piquemal@sorbonne-universite.fr; pierre.monmarche@univ-eiffel.fr

possible to construct a nonreversible chain that converges faster than the reversible counterpart to the same stationary distribution[18,19]. One way to do this is to use a so-called lifting procedure. An optimal lift can offer up to a quadratic speedup (like the quantum algorithms)[20,21] but typically requires a detailed knowledge of the chain[20]. Because there is no downside to using lifts, nonreversible processes are generally preferred for sampling purposes[22]. Nonreversible processes may also be used to encode reversible ones, such as Metropolis-Hastings kernels, more efficiently[23]. Because of the growing interest in such processes, we ask the following question.

Can quantum algorithms accelerate the mixing of nonreversible Markov processes?

In summary, we answer this question by:
- analyzing known quantum singular value transforms in the context of nonreversible Markov chains (which requires simulating the time-reversal), and
- introducing quantum eigenvalue transforms to retrieve an approximation of the stationary distribution without simulating the time-reversal.

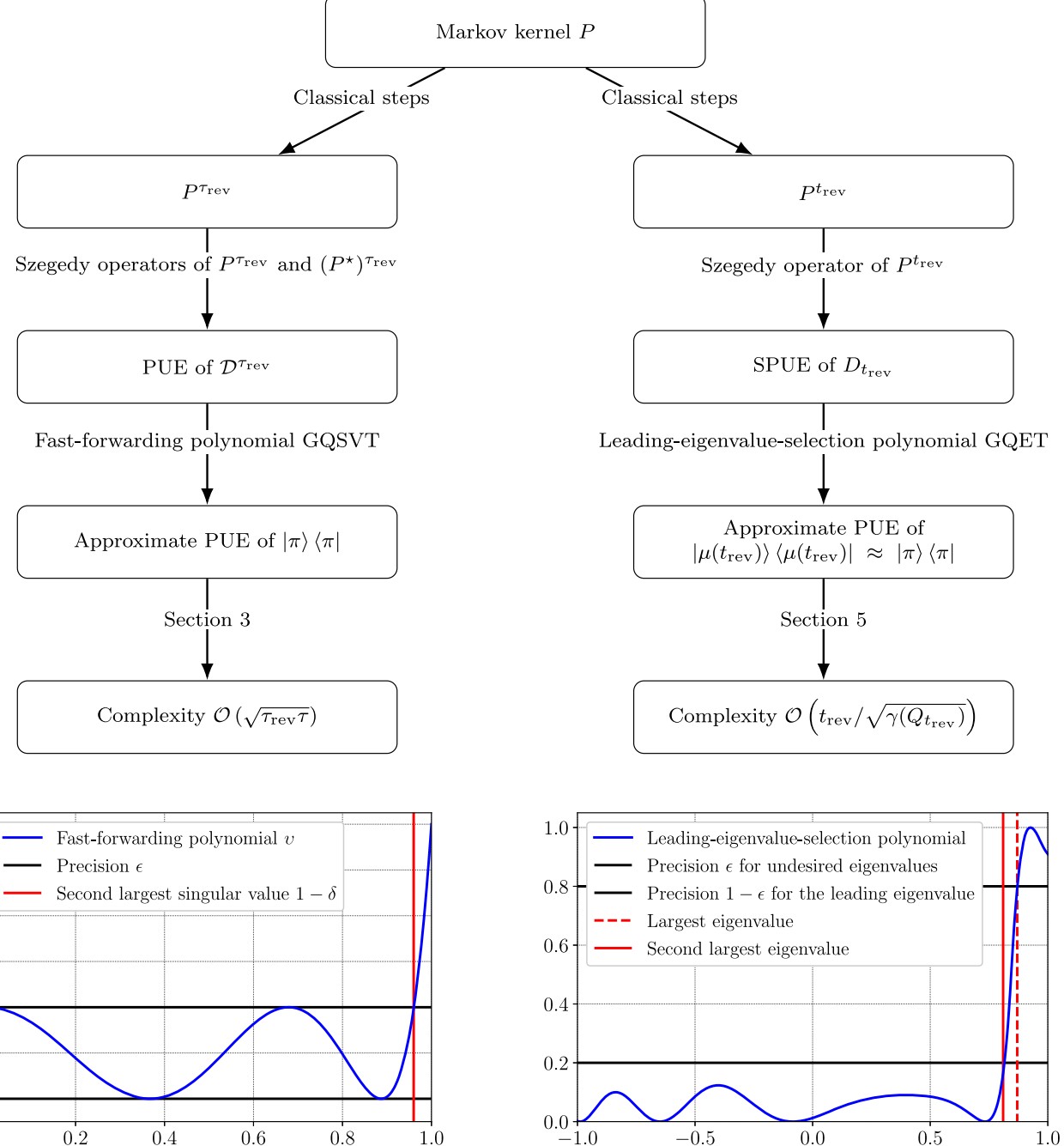

**Fig. 1 | Algorithmic workflow for the curved and flat discriminant approaches.** Both algorithms start by taking classical steps of the Markov chain until a condition is met. Either through singular value or eigenvalue transform, a polynomial is applied to the corresponding discriminant in order to obtain a projected unitary encoding of the sought operator $|\pi\rangle\langle\pi|$. (S)PUE denotes (Symmetric) Projected Unitary Encodings, GQSVT stands for Generalized Quantum Singular Value Transform, and GQET for Generalized Quantum Eigenvalue Transform.

More precisely, this work provides two ways to construct an approximate reflection through the target coherent state from the Szegedy quantum walk operators[11] (section "Encoding Markov chains in quantum computers" provides further details on the oracle access model). Its main contributions are the algorithmic workflows presented in Fig. 1. In both methods, we perform classical steps of the walk until a certain criterion is met and the chain is "reversible enough." Then, we perform a suitable quantum singular value or eigenvalue transform, producing the desired reflection with a speedup. Amplitude amplification algorithms use such a reflection to prepare the sought state[24]. Since QAE relies on such a state preparation unitary[4,5], the reflection is readily usable in Quantum Monte Carlo (QMC) routines[3]. QMC requires quadratically fewer samples from the stationary distribution than classical MCMC, and each sample can be provided more efficiently.

In the first method, the number of steps to be performed classically is called the reversibilization time $\tau_{\mathrm{rev}}$ and has a precise definition. The algorithm constructs the desired reflection in a time of the order of $\sqrt{\tau_{\mathrm{rev}}\tau}$, where $\tau$ is the mixing time of the chain (see the section "Singular value transformations of non-normal Markov kernels"). The second method builds on the same intuition to define the number $t_{\mathrm{rev}}$ of steps to be performed classically before the quantum eigenvalue transformation is applied (see Fig. 1). We introduce a Markov kernel called the geometric reversibilization of the chain and a condition of reversibility on $\pi$-average. This condition guarantees both the efficiency of the method and the quality of the produced reflection. Indeed, if this condition is met, the geometric reversibilization reaches its stationary distribution in a time of the order of the mixing time of the additive reversibilization of the chain, the process that goes forward and backward in time with equal probabilities. This stationary distribution is called the most reversible distribution and is guaranteed to be close to $\pi$. Figure 2 illustrates an example where the condition of reversibility on $\pi$-average is verified long before the mixing time. Figure 2a displays that the eigenvalues of the geometric reversibilization, whose properties govern the runtime of the quantum algorithm, rapidly approach those of the additive reversibilization of the kernel. Figure 2b shows that this phenomenon occurs long before the mixing time of the kernel, resulting in a runtime of the square root of the mixing time of the additive reversibilization (see the section "Reversibility on π-average and another notion of reversibilization time").

By efficiently constructing reflections through the stationary distribution of nonreversible chains, this study expands the set of Markov chain computations that can be achieved faster using a quantum computer. These reflections could ultimately be applied to accelerate molecular dynamics, with implications in drug design[25,26], in protein folding studies[27–30] and in any subfield using molecular simulations. They could also help simulating the limiting behavior of stochastic differential equations, with implications in financial modeling[31,32].

## Results

### Technical background

In this section, we first summarize Markov chain concepts[17]. We then describe core quantum algorithmic tools such as projected unitary encodings and qubitized walk operators, more detailed expositions of which can be found in refs. 33,34. We observe that, in order to get a reflection through the stationary distribution, it is sufficient to obtain a projected unitary encoding of the projector on this state. We express the latter as polynomials of discriminant matrices, which are related to Markov kernels and can be encoded through projected unitary encodings. Finally, we recall how the Generalized Quantum Eigenvalue Transform (GQET) and the Generalized Quantum Singular Value Transform (GQSVT) allow us to construct a projected unitary encoding of such polynomials.

**Markov chains and mixing time.** A Markov kernel on a finite set $\mathbb{S}$ is a matrix $P$ of positive numbers such that each row sums to 1. The Markov chain with initial condition $X_0 \in \mathbb{S}$ associated with $P$ is a stochastic process $(X_t)_{t \in \mathbb{N}}$ with values in $\mathbb{S}$ such that when it is in state $x \in \mathbb{S}$, it goes to $y \in \mathbb{S}$ with probability $P(x, y)$. If the kernel allows reaching any state $y \in \mathbb{S}$ from any state $x \in \mathbb{S}$, we say that it is irreducible. If it has a single eigenvalue on the unit circle, we say that it is aperiodic. If $P$ is both irreducible and aperiodic, it is said to be ergodic. An ergodic Markov kernel is such that, after waiting for a sufficiently long time $\tau \in \mathbb{N}$, the probability distribution of the system state $X_\tau$ is approximately given by a probability distribution $\pi$ that is independent of the starting state (see the Methods "Markov chains" for a precise definition of the mixing time $\tau$). This distribution is called the stationary distribution. A sufficient condition for an ergodic kernel $P$ to have a stationary distribution $\pi$ is to be reversible with respect to $\pi$: such that $\pi(x)P(x, y) = \pi(y)P(y, x)$ for any states $x, y \in \mathbb{S}$. Defining the time-reversal $P^\star$ by $P^\star(x, y) = \pi(y)P(y, x)/\pi(x)$ for every $x, y \in \mathbb{S}$, the previous condition can be rewritten $P = P^\star$. We will refer to $(P + P^\star)/2$ as the additive reversibilization of $P$, and to $PP^\star$ as the multiplicative

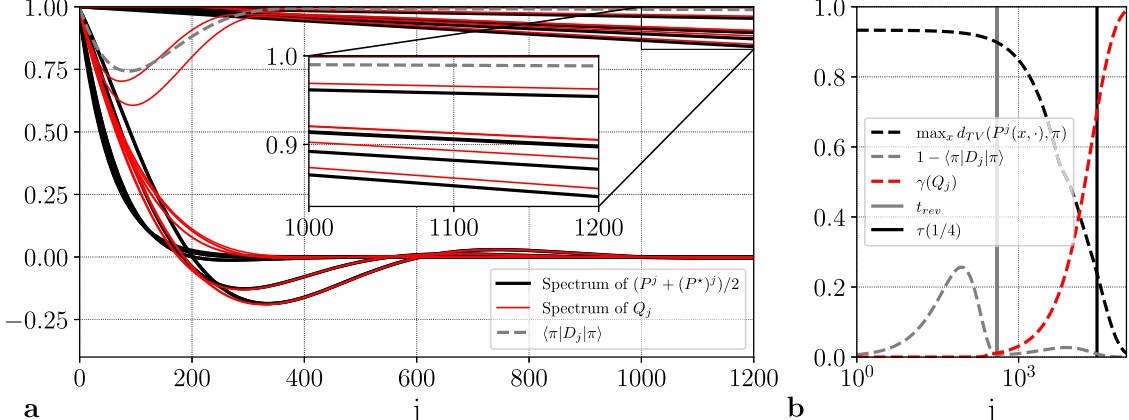

**Fig. 2 | Geometric reversibilizations and reversibility on $\pi$-average. a** The relationship between the spectra of the additive and geometric reversibilizations of a Markov kernel $P^j$, as a function of the number of Markov chain steps $j$. The additive reversibilization $(P^j + (P^\star)^j)/2$ is the process that goes forward or backward in time with equal probabilities. The geometric reversibilization $Q_j$ is the process whose spectral gap $\gamma(Q_j)$ governs the complexity of the quantum algorithm. **b** The condition of reversibility on $\pi$-average, $1 - \langle\pi|D_j|\pi\rangle \ll \gamma(Q_j)$, is verified long before the mixing time $\tau(1/4)$. The quantum algorithm does provide a speedup if this condition is satisfied.

reversibilization of $P$. MCMC algorithms compute the expectation values of functions of random variables sampled according to $\pi$. Therefore, the number of Markov chain steps required to provide a sample from $\pi$ is a natural performance metric.

**Mixing times and notions of spectral gap.** The mixing time of ergodic reversible Markov kernels is related to their spectral gaps. If $P$ is an ergodic reversible Markov kernel, define its spectral gap to be $\gamma(P) = 1 - \max_{\lambda \in \sigma(P) \setminus \{1\}} |\lambda|$. Then, the mixing time of $P$ is essentially of the order of $1/\gamma(P)$[17]. As it turns out, this relationship does not hold true for general Markov kernels. If $P$ is an ergodic kernel, we may define its pseudo-spectral gap $\gamma_\infty(P)$ by $\gamma_\infty(P) = \max_{k \geq 1} \gamma\left(P^k (P^\star)^k\right)/k$[35]. Then, the previous relationship can be generalized in the sense that the mixing time of $P$ is of the order of $1/\gamma_\infty(P)$.

**Projected unitary encodings and qubitized walk operators.** A unitary $U$ and two (partial) isometries $\square_L$, $\square_R$ are said to be a Projected Unitary Encoding (PUE) $(U, \square_L, \square_R)$ of $A$ if $\square_L^\dagger U \square_R = A$ (the superscript $\dagger$ indicates the conjugate transpose matrix). If $\square_L = \square_R = \square$ and $U$ is also symmetric, then $(U, \square)$ is said to be a Symmetric Unitary Projected Encoding (SPUE) of $A$. Given $(U, \square)$ a SPUE of $A$, define the qubitized walk operator $\mathcal{W} = (2\square\square^\dagger - 1)U$. This operator is going to be particularly useful when $A$ is the projector on a state of interest. Indeed, if $A = |\phi\rangle\langle\phi|$ for some state $|\phi\rangle$, then $(\mathcal{W}^2, \square)$ is a SPUE of $2|\phi\rangle\langle\phi| - 1$ (see equation (12) in the "Methods").

**Encoding Markov chains in quantum computers.** Let us now define PUEs, namely unitary operators and partial isometries, that encode information about Markov kernels. Consider the Hilbert space spanned by the orthonormal computational basis $\{|x, y\rangle\}_{x, y \in \mathbb{S}}$ with the register swap operator $S = \sum_{x, y \in \mathbb{S}} |x, y\rangle\langle y, x|$. For each Markov kernel $P : \mathbb{S}^2 \to [0, 1]$, define the partial isometry $\square_P = \sum_{x \in \mathbb{S}} |x\rangle |P(x, \cdot)\rangle \langle x|$ (for every probability distribution $\eta$ on $\mathbb{S}$, we write $|\eta\rangle = \sum_{x \in \mathbb{S}} \sqrt{\eta(x)}|x\rangle$). If $P_1, P_2 : \mathbb{S}^2 \to [0, 1]$ are two Markov kernels, then for each $x, y \in \mathbb{S}$:

$$\left\langle x \left| \square_{P_1}^\dagger S \square_{P_2} \right| y \right\rangle = \sqrt{P_1(x, y) P_2(y, x)}. \tag{1}$$

In words, $(S, \square_{P_1}, \square_{P_2})$ is a PUE of the matrix $D_{P_1, P_2} : \mathbb{S}^2 \to [0, 1]$, $(x, y) \mapsto \sqrt{P_1(x, y) P_2(y, x)}$. We will be particularly interested in the so-called flat discriminant $D = D_{P,P}$, with SPUE $(S, \square) = (S, \square_P)$, and curved discriminant $\mathcal{D} = D_{P, P^\star}$, with PUE $(S, \square^\star, \square) = (S, \square_{P^\star}, \square_P)$. We will be interested in cases where one of the eigenvectors (resp. singular vector) of $D$ (resp. $\mathcal{D}$) is a good approximation $|\mu\rangle$ for $|\pi\rangle$. Then, we may write our target projector $|\mu\rangle\langle\mu| = \upsilon(D)$, where $\upsilon$ is a polynomial such that $\upsilon(\mu) = 1$ and $\upsilon(\lambda) = 0$ for other eigenvalues $\lambda \neq \mu$ of $D$. Constructing encodings of such $\upsilon(D)$ is precisely the aim of the GQET and GQSVT presented below. It will make use of the Szegedy quantum walk operators[11], $R = 2\square\square^\dagger - 1$ and $R^\star = 2\square^\star(\square^\star)^\dagger - 1$. The operators $R$ and $R^\star$ are typically constructed with arithmetic oracles capable of performing the transformation

$$|x\rangle \to \sum_{y \in \mathbb{S}} \sqrt{P(x, y)}|x, y\rangle, \tag{2}$$

for every state $x \in \mathbb{S}$ (see, for example, ref. 36). Such oracles are efficiently implementable whenever the transition probabilities are efficiently computable. If the stationary distribution $\pi$ can be computed up to a multiplicative constant, ratios of the form $\pi(y)/\pi(x)$ can be efficiently evaluated for any pair of states $x, y \in \mathbb{S}$. Consequently, the time-reversed transition probabilities $P^\star(x, y) = P(y, x)\pi(y)/\pi(x)$ can also be computed efficiently, implying that $P^\star$ is accessible under the same oracle model. As demonstrated in refs. 37–39 and exemplified in the Supplementary Information, it is often possible to construct these

operators much more efficiently. For kinetic processes, where the state space includes both position and velocity components (as in underdamped Langevin dynamics), $P^\star$ can be efficiently implemented by reversing the initial velocity, applying one step of $P$, and then reversing the velocity again.

**Generalized Quantum Eigenvalue Transform.** Let $(U, \square)$ be a SPUE of an operator $H$. Let $\upsilon$ be a polynomial of degree $d$ with complex coefficients. The GQET is a unitary $\mathcal{U}$ built from $d$ uses of controlled-$\mathcal{W}$ operations such that $(\mathcal{U}, |0\rangle \otimes \square, |0\rangle \otimes \square)$ is a PUE of $\upsilon(H)/\beta$. $\mathcal{W}$ denotes the qubitized walk operator of $(U, \square)$ and $1 \leq \beta \in \mathcal{O}(\log(d))$ is called the scaling factor of $\upsilon$[34].

**Generalized Quantum Singular Value Transform.** Let $(U, \square_L, \square_R)$ be a PUE of an operator $A$ and $\upsilon$ be an even polynomial of degree $d$ with complex coefficients. The GQSVT is a unitary $\mathcal{U}$ built from $d$ controlled-$U$, controlled-$U^\dagger$, controlled-$(2\square_L \square_L^\dagger - 1)$ and controlled-$(2\square_R \square_R^\dagger - 1)$ such that $(\mathcal{U}, |01\rangle \otimes \square_R, |01\rangle \otimes \square_R)$ a PUE of $V^\dagger \upsilon(\Sigma)V/\beta$, where $A = W^\dagger \Sigma V$ is the singular value decomposition of $A$ and $\beta$ is the same scaling factor as in GQET.

**Applying the fast-forwarding polynomial.** Let $(S, \square^\star, \square)$ be a PUE of $\mathcal{D}$, the curved discriminant of an ergodic Markov kernel. Without loss of generality, we assume that the singular values of $\mathcal{D}$ are in $[0, 1-\delta] \cup \{1\}$, for some $\delta \in ]0, 1[$, with unique left and right singular vectors with singular value 1 denoted by $|\pi\rangle$. We want to apply the GQSVT with a polynomial $\upsilon$ of low degree such that $V^\dagger \upsilon(\Sigma)V/\beta$ is an approximation of $|\pi\rangle\langle\pi|$, where $\mathcal{D} = W^\dagger \Sigma V$ is the singular value decomposition of $\mathcal{D}$. As detailed in the Methods "Quadratic speedup polynomial", there exists an implementable polynomial $\upsilon$ of degree $\mathcal{O}\left(\delta^{-1/2} \log(1/\epsilon)\right)$, and scaling factor $\beta = 1$, such that $\left\|V^\dagger \upsilon(\Sigma)V/\beta - |\pi\rangle\langle\pi|\right\| \leq \epsilon$. An example of such a polynomial $\upsilon_{8, 0.2}(x)$ of degree 8 is illustrated in Fig. 1, with $\epsilon = 0.2$. The polynomial is more efficient than the monomial $x^8$, corresponding to the execution of 8 steps of the chain, in the sense that the preimage of $[-\epsilon, \epsilon]$ is a much wider interval. For this reason, we will refer to $\upsilon$ as the fast-forwarding polynomial.

### Singular value transformations of non-normal Markov kernels

The GQSVT algorithm allows to apply fixed-parity polynomials to the singular values of $P$, the square roots of the eigenvalues of $PP^\star$. Therefore, the performances of the algorithm strongly depend on the mixing time of the multiplicative reversibilization $PP^\star$ of $P$. However, up to considering the lazy version of $P$, $PP^\star$ may mix quadratically slower than $P$[18]. The quantum speedup provided by the GQSVT transformation being quadratic, there are no benefits to directly transforming the singular values of $P$. In order to optimize the procedure, we suggest to apply the GQSVT algorithm to $P^k$, for a number of steps $k$ left unspecified, and at the expense of multiplying the cost of the method by $k$. Is there an integer $k$ such that the resulting construction requires much fewer steps than the mixing time?

As described in the "Technical background" section "Markov chains and mixing time", the mixing time is related to the inverse of the pseudo-spectral gap. By definition of the pseudo-spectral gap, there exists a smallest integer $\tau_{rev} \geq 1$ such that $\gamma_\infty(P) = \gamma\left(P^{\tau_{rev}}(P^\star)^{\tau_{rev}}\right)/\tau_{rev}$. Applying the GQSVT transformation to $P^{\tau_{rev}}$ to encode an approximation of $|\pi\rangle\langle\pi|$ with $\mathcal{O}\left(1/\sqrt{\gamma\left(P^{\tau_{rev}}(P^\star)^{\tau_{rev}}\right)}\right)$ uses the PUE of $P^{\tau_{rev}}$. Thus, the overall procedure has complexity:

$$\mathcal{O}\left(\frac{\tau_{rev}}{\sqrt{\gamma\left(P^{\tau_{rev}}(P^\star)^{\tau_{rev}}\right)}}\right) = \mathcal{O}\left(\sqrt{\frac{\tau_{rev}}{\gamma_\infty(P)}}\right). \tag{3}$$

Recalling that $1/\gamma_\infty(P)$ is upper bounded by the mixing time, the overall complexity is in:

$$\mathcal{O}\left(\sqrt{\tau_{rev}\tau}\right). \tag{4}$$

Proposition 1 summarizes the derivation.

**Proposition 1.** Let $P$ be an ergodic kernel on finite state space $\mathbb{S}$. A quantum circuit approximating $2|\pi\rangle\langle\pi| - 1$ up to spectral norm error $\epsilon > 0$ can be constructed with $\mathcal{O}\left(\sqrt{\tau_{rev}\tau(\epsilon)}\log(1/\epsilon)\right)$ uses of the Szegedy quantum walk operators $R$ and $R^\star$, where $\tau_{rev}$ is the reversibilization time of $P$, and $\tau(\epsilon)$ its $\epsilon$-mixing time.

To summarize, we may reflect through the stationary measure of a Markov process with speedup whenever the reversibilization time of the process is smaller than its mixing time.

### Eigenvalue transformations of the flat discriminant

In practice, we may not be able to access $R^\star$. Indeed, the time-reversal of the chain may not be easy to sample from[37] or the stationary distribution may not be known[16]. Let us discuss how to implement $2|\pi\rangle\langle\pi| - 1$ using only $R$. Recall that $(S, \square)$ is a SPUE of the flat discriminant $D$, a symmetric matrix with eigenvalues in $[-1, 1]$.

Let us now construct a new reversibilization of a Markov kernel. If $D$ is primitive, meaning that $D^m > 0$ element-wise for some $m \in \mathbb{N}$, then the Perron-Frobenius Theorem (Theorem 6 in the Supplementary Information) and the variational principle imply the existence of a strictly positive probability distribution $\mu$ on $\mathbb{S}$ such that:

$$\mu = \text{argmax}_\nu \langle\nu|D|\nu\rangle, \tag{5}$$

where the argmax ranges over all strictly positive probability distributions on $\mathbb{S}$. The geometric reversibilization of $P$, denoted by $Q : \mathbb{S}^2 \to [0, 1]$, and defined by:

$$\forall x, y \in \mathbb{S} : Q(x, y) = \sqrt{\frac{\mu(y)}{\mu(x)}} \frac{D(x, y)}{\langle\mu|D|\mu\rangle} \tag{6}$$

is well-defined. Moreover, $Q$ is a Markov kernel that is reversible with respect to $\mu$. $D$ being primitive, it is in fact ergodic with a unique stationary distribution $\mu$. Also, the spectra of $D$ and $Q$ are related through the equation $\langle\mu|D|\mu\rangle\sigma(Q) = \sigma(D)$. According to the variational principle, $\langle\pi|D|\pi\rangle$ is a lower bound for the largest eigenvalue of $D$. Moreover, if $\langle\pi|D|\pi\rangle$ is much closer to 1 than to $1-\gamma(Q)$, where $\gamma(Q)$ is the spectral gap of $Q$, then the overlap $\langle\pi|\mu\rangle$ between the stationary distribution and the most reversible distribution is large. We refer to this condition as reversibility on $\pi$-average. If a kernel is reversible on $\pi$-average, then we are interested in encoding the reflection $2|\mu\rangle\langle\mu| - 1$. In order to implement this reflection up to spectral norm error $\epsilon$ using the GQET formalism, we need a polynomial $\upsilon$ of low degree such that $\upsilon(\langle\mu|D|\mu\rangle) \geq 1-\epsilon$ and $|\upsilon(x)| \leq \epsilon$ for all $x \in [-1, \langle\mu|D|\mu\rangle(1 - \gamma(Q))]$. What is the minimal degree for a polynomial satisfying this property?

Let us construct such a polynomial as the composition of $\upsilon$, which was applied to the singular values of the curved discriminant (displayed in Fig. 1), with another polynomial. If $P$ is a nonreversible Markov chain, $\upsilon(\langle\mu|D|\mu\rangle) < 1$. We compose $\upsilon$ with a function $f : [-1, 1] \to \mathbb{R}$ such that $f(x) = 1$ for all $x \in [\langle\mu|D|\mu\rangle, 1]$ and $f(x) = 0$ for all $x \in [-1, \langle\mu|D|\mu\rangle[$. Applying $f \circ \upsilon$ to the eigenvalues of the flat discriminant would construct a PUE of $|\mu\rangle\langle\mu|$ from which we can recover $2|\mu\rangle\langle\mu| - 1$. $f$ is not a polynomial but can be efficiently approximated by a polynomial, called a leading-eigenvalue-selection polynomial. The resulting composite polynomial is illustrated in Fig. 1. It is of low degree if the gap between the two leading eigenvalues of $D$ does not close more rapidly than the gap between the first eigenvalue and 1, as stated by

Proposition 2. A proof is given in the "Methods" section "Quadratic speedup polynomial".

**Proposition 2.** Let $0 < \delta < 1$ and $\epsilon > 0$. Then, there exists a constant $c \in ]0,1[$ and a real polynomial $\upsilon$ of degree $\mathcal{O}\left(\delta^{-1/2}\log(1/\epsilon)\right)$ such that:
- $|\upsilon(x)| \leq 1$ for all $x \in [-1, 1]$,
- $|\upsilon(x)| \leq \epsilon$ for all $x \in [-1, 1-\delta]$ and
- $\upsilon(x) \geq 1-\epsilon$ for all $x \in [1-c\delta, 1]$.

If $Q$ has a mixing time that is small compared to $1/(1 - \langle\pi|D|\pi\rangle)$, Proposition 2 provides a polynomial $q$ such that $q(D)$ is a projection on $|\mu\rangle$ and ensures a large overlap $\langle\pi|\mu\rangle$. The result is summarized in Proposition 3 (see the Methods "Quadratic speedup polynomial" for a proof).

**Proposition 3.** Let $P$ be an ergodic kernel on finite state space $\mathbb{S}$ with primitive flat discriminant, most reversible distribution $\mu$, and geometric reversibilization $Q$ with spectral gap $\gamma(Q)$. Let $c \in ]0,1[$ be the constant introduced in Proposition 2. If

$$\frac{1-c}{1-c(1-\gamma(Q))} \leq \langle\mu|D|\mu\rangle, \tag{7}$$

then $2|\mu\rangle\langle\mu| - 1$ can be prepared up to spectral norm $\epsilon > 0$ with $\tilde{\mathcal{O}}\left(\gamma(Q)^{-1/2}\log(1/\epsilon)\right)$ uses of the Szegedy quantum walk operator $R$. Moreover,

$$\langle\mu|\pi\rangle^2 \geq 1 - \frac{\langle\mu|D|\mu\rangle - \langle\pi|D|\pi\rangle}{\langle\mu|D|\mu\rangle\gamma(Q)}. \tag{8}$$

### Reversibility on $\pi$-average and another notion of reversibilization time

If $P$ is reversible, then $D = \mathcal{D}$ is equal to $P$ up to a change of basis. If $P$ is nonreversible, then $D$ may contain no information about the stationary distribution. For example, consider a Markov chain on the $N$-point discrete circle that goes clockwise with probability 1/2 and stays where it is otherwise: $D$ is half the identity matrix. Indeed, for every $x \neq y$, $P(x, y) > 0$ implies $P(y, x) = 0$ and therefore $D(x, y) = \sqrt{P(x, y)P(y, x)} = 0$. However, if $P(x, y) = \pi(y)$ for each $x, y \in \mathbb{S}$, meaning that it is perfectly mixed, then $D = |\pi\rangle\langle\pi|$ and $R = 2\sum_{x,y,z\in\mathbb{S}}\sqrt{\pi(y)\pi(z)}|x, y\rangle\langle x, z| - 1 = 1 \otimes (2|\pi\rangle\langle\pi| - 1)$. As a consequence, it is much more appropriate to study the sequence $(D_j)_{j\in\mathbb{N}}$ of the flat discriminants associated to $\left(P^j\right)_{j\in\mathbb{N}}$ instead of simply $D = D_1$. Indeed, we found that applying GQSVT to $P^{\tau_{rev}}$ instead of $P$ improved the efficiency of the reflection construction. We want to build on this intuition and apply a GQET to the flat discriminant of $P^j$, for some integer $j \geq 1$. In light of Proposition 3, we will choose $j$ to be the least integer such that $P^j$ is reversible on $\pi$-average: such that $1 - \langle\pi|D_j|\pi\rangle \ll \gamma(Q_j)$. Because it plays a role similar to that of $\tau_{rev}$, we will denote this integer $j$ by $t_{rev}$. The remainder of the section will describe situations where $t_{rev}$ is much smaller than the mixing time. In this case, applying the GQET to $D_{t_{rev}}$ leads to both a good approximation of the desired reflection and presents a complexity that is smaller than the mixing time of the chain.

Figure 3 illustrates the phenomenon for a nonreversible walk on a graph with bottleneck. We will consider two non-symmetric (therefore nonreversible) walks on two circles connected by a bridge that is only taken with small probability. Let $N$ be odd ($N = 31$ for the numerical experiment) and consider the state space $\mathbb{S} = \{0, ..., N-1\} \cup \{\partial\} \cup \{N, ..., 2N-1\}$. Define the nonreversible

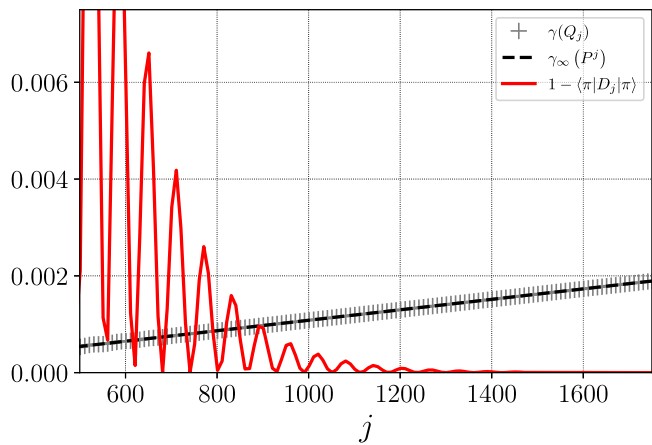

**Fig. 3 | Nonreversible walk on a graph with bottleneck.** The condition of reversibility on $\pi$-average $1 - \langle \pi | D_j | \pi \rangle \ll \gamma(Q_j)$ is verified for $j \in \mathbb{N}$ much smaller than the mixing time. As a consequence, $\gamma(Q_j)$ is of the same order as $\gamma_\infty(P^j)$, the pseudo-spectral gap of $P^j$.

Markov chain $P$ by:

$$\begin{aligned} P(x, (x+1)[N]) &= P(N+x, N+(x+1)[N]) &= 3/4, \\ P(x, (x-1)[N]) &= P(N+x, N+(x-1)[N]) &= 1/4, \end{aligned} \tag{9}$$

for all $x \in \{1, \ldots, N-1\}$, where $[N]$ denotes modulo $N$ and

$$\begin{cases} P(0, \partial) &= P(N, \partial) &= 1/N^3, \\ P(0, 1) &= P(N, N+1) &= (1 - 1/N^3)3/4, \\ P(0, N-1) &= P(N, 2N-1) &= (1 - 1/N^3)/4, \\ P(\partial, 0) &= P(\partial, N) &= 1/2. \end{cases} \tag{10}$$

If the process starts in $\{0, N-1\}$ and does not exit this set, then it converges to a quasi-stationary distribution in $\mathcal{O}(N^2)$ steps. The same holds if the starting set is $\{N, 2N-1\}$. Since it takes of the order of $N^3$ steps to exit the set, the overall mixing time is at least of order $N^3$. Figure 3 shows that the fast convergence to a quasi-stationary distribution is sufficient for $D$ to present the desired spectral properties. In particular, the figure shows that $1/\sqrt{\gamma(Q_j)}$ is of the order of the square root of the mixing time of $P^j$.

Note that $\langle \pi | D_j | \pi \rangle$ is the $\pi$-averaged overlap between the laws $P^j(x, \cdot)$ and $(P^*)^j(x, \cdot)$ as $x$ follows $\pi$: $\langle \pi | D_j | \pi \rangle = \mathbb{E}_\pi[\langle P^j(x, \cdot) | (P^*)^j(x, \cdot) \rangle]$. In practice, a chain with local moves tends to stay for a long time near a local maxima of probability (according to the distribution $\pi$), regardless of whether it evolves according to $P$ or $P^*$. In such regions, the overlap between the laws of $P$ and of $P^*$ tends to 1 very fast (1 minus the overlap is in fact the square of a distance). For $x \in \mathbb{S}$ in a low probability region, $P^j(x, \cdot)$ and $(P^*)^j(x, \cdot)$ can remain different until $j$ equals the mixing time; e.g., kinetic processes and their time-reversal tend to leave local minima of probabilities in opposite directions. Since such initial conditions are of low probability, their contribution to the mean is of low importance and $\langle \pi | D_j | \pi \rangle = \mathbb{E}_\pi[\langle P^j(x, \cdot) | (P^*)^j(x, \cdot) \rangle]$ approaches 1. Such ideas are made rigorous using the concept of quasi-stationary distributions in Proposition 8. A numerical example is given in the Supplementary Information. Many discrete approximations of stochastic differential equations also typically have quasi-stationary distributions and rapidly increasing $\langle \pi | D_j | \pi \rangle$ parameter. More details regarding such processes are given in the Supplementary Information.

Note that the flat discriminant of a Markov kernel is equal to the flat discriminant of its adjoint. Moreover, a kernel and its adjoint do not have the same mixing time in general. Corollary 2, given in the "Methods" section "Mixed kernels" and proven in the Supplementary Information, shows that if either a kernel or its adjoint is sufficiently mixed, then the reflection through the most stationary distribution can

be constructed with constant cost. In the Supplementary Information, we give an example of a Markov chain such that the reflection construction time is exponentially smaller than the mixing time.

Let us now assume that the Markov chain possesses a group structure, then $\|D_j - (\mathcal{D}^j + (\mathcal{D}^\dagger)^j)/2\| = 1 - \langle \pi | D_j | \pi \rangle$. This implies that the eigenvalues of the flat discriminant are all within distance $1 - \langle \pi | D_j | \pi \rangle$ of an eigenvalue of $(P^j + (P^*)^j)/2$. A precise statement is given in Proposition 7. For many applications, it is therefore sufficient to focus on $\langle \pi | D_j | \pi \rangle$.

## Discussion
### Summary
Let us now come back to our initial question: can quantum algorithms accelerate the mixing of nonreversible Markov processes?

We started by analyzing the performance of the GQSVT algorithm when applied to a nonreversible kernel $P$. The algorithm runs in the square root of the mixing time of $PP^*$, which has a quadratically better worst-case dependency on the spectral gap $\gamma$ than the mixing time. We then noticed that applying the procedure to $P^j$ instead of $P$, thus multiplying the oracle cost by $j$, could improve the overall complexity. This is of particular interest when the optimal $j$ is much smaller than the mixing time.

Without requiring the use of the Szegedy quantum walk associated with the time-reversal of the kernel, it is still possible to encode the flat discriminant of the chain. We introduced a notion of reversibility on $\pi$-average. We constructed a polynomial of low degree (i.e., square root of the inverse gap) such that, when the condition is verified, it allows us to approximately encode the desired projector on the stationary distribution through the GQET. If $P^j$ is reversible on $\pi$-average, for a $j$ significantly smaller than the mixing time, then the quantum algorithm provides a speedup for sampling according to $\pi$.

We therefore provided sufficient conditions for quantum algorithms to accelerate sampling from the distribution of nonreversible kernels. Since both classical and quantum algorithms present complexity lower bounds of the order of the diameter of the underlying Markov kernel, there is no hope to accelerate processes that mix in the time required to cross their underlying graph. Processes that achieve this lower bound mix efficiently. Such processes are hard to design classically and are not representative of practical cases. We show that, when the quadratic relationship between the quantum and classical runtime is broken, the speedup in sampling from $\pi$ can be more than quadratic. However, when studying the sequences of discriminants associated with different kernels, several different techniques came into play. Specifically, we could not provide a general, easy way to check the reversibility on the $\pi$-average condition. This issue may be solved experimentally by estimating the phases of the qubitized walk operators in order to obtain the missing information about the spectra of the discriminants.

### Research directions
One of the key theoretical challenges when using MCMC algorithms, in particular those using nonreversible Markov chains, is estimating the mixing times. This issue is usually addressed on a case-by-case basis by practitioners. Here, we list several research directions to pursue the present work towards applications.

In the computational study of statistical physics systems, such as in molecular simulations, reversible Markov kernels are a central tool. However, such processes have sometimes failed to provide good results due to their slow exploration of the configuration space. Significant effort is devoted to replacing local reversible processes with local nonreversible ones that could explore the space more rapidly[25]. While still in its infancy, event-chain Monte Carlo uses nonreversible processes and could accelerate molecular simulation of proteins in aqueous solution.

Much effort is also currently being devoted to the development of mathematical models for biochemical networks[40]. Biological processes appear to be best described by a mixture of stochastic, continuous, and discrete phenomena. Thus, the appropriate mathematical object to employ is that of a nonreversible piecewise deterministic Markov process.

In the context of financial modeling, reversible diffusions appear to poorly describe sudden price moves[32]. Nonreversible jump processes seem to model such moves much more realistically. A practitioner might therefore be interested in studying the long-term behavior of a certain model, without a priori knowledge of its stationary distribution.

Further effort could also be devoted to tighter estimates for the spectra of flat discriminants or the efficient construction of the Szegedy walk operators for particular problems.

## Methods
### Markov chains
Let us start by recalling central definitions. Let $\mu, \nu : \mathbb{S} \to [0,1]$ be such that $\sum_{x \in \mathbb{S}} \mu(x) = \sum_{x \in \mathbb{S}} \nu(x) = 1$. The total variation distance separating them is defined as $d_{TV}(\mu, \nu) = \frac{1}{2} \sum_{x \in \mathbb{S}} |\mu(x) - \nu(x)|$. Let $\epsilon > 0$. The $\epsilon$-mixing time of an ergodic Markov kernel $P$ with stationary distribution $\pi$ is defined by $\tau(\epsilon) = \min\{t \in \mathbb{N} : \max_{x \in \mathbb{S}} d_{TV}(P^t(x, \cdot), \pi) \leq \epsilon\}$. The default precision is $\epsilon = 1/4$ so that the notation $\tau$ means $\tau = \tau(1/4)$. A kernel $P$ is said to be lazy if $P(x, x) \geq 1/2$ for each $x \in \mathbb{S}$. The mixing time of $P$ is related to its pseudo-spectral gap $\gamma_\infty(P)$ through the inequalities:

$$\frac{1 - \epsilon}{\gamma_\infty(P)} \leq \tau(\epsilon) \leq \frac{1 - \ln(2\epsilon\pi_*)}{\gamma_\infty(P)}, \tag{11}$$

where $\pi_* = \min_{x \in \mathbb{S}} \pi(x)$.

### PUE of projectors and reflections
Let us explain why it is sufficient to construct the SPUE of $|\pi\rangle\langle\pi|$. Assume that $(U, \square)$ is an SPUE of $A = |\phi\rangle\langle\phi|$, for a normalized state $|\phi\rangle$. Recall the definition of the qubitized walk operator $\mathcal{W} = (2\square\square^\dagger - 1)U$. Compute:

$$\begin{aligned}
\square^\dagger \mathcal{W}^2 \square &= \square^\dagger U(2\square\square^\dagger - 1)U\square \\
&= 2\square^\dagger U\square\square^\dagger U\square - \square^\dagger U^2\square \\
&= 2|\phi\rangle\langle\phi|\phi\rangle\langle\phi| - \square^\dagger\square \\
&= 2|\phi\rangle\langle\phi| - 1.
\end{aligned} \tag{12}$$

Since $\square$ is a partial isometry, $\square^\dagger\square$ is the projection on its support. Thus, $\square^\dagger \mathcal{W}^2 \square$ acts as $2|\phi\rangle\langle\phi| - 1$ on the support of $\square$.

### Quantum signal processing
We can now describe precisely the quantum signal processing tools at our disposal. The central result provides PUEs of polynomials of unitaries, and is summarized in Theorem 1 (see ref. [33]).

**Theorem 1.** Let $\Upsilon$ be a degree $d \in \mathbb{N}$ complex polynomial and $U$ be a unitary. If $\|\Upsilon(V)\| \leq 1$ for all unitaries $V$, then we can construct a PUE $(W, |0\rangle, |0\rangle)$ of $\Upsilon(U)$ using $d$ controlled-$U$ operations, $\mathcal{O}(d)$ additional single-qubit gates and 1 ancilla qubit.

Theorem 1 can be used to apply polynomials to non-unitary symmetric matrices. Theorem 2 shows how to construct PUEs of a polynomial of a symmetric matrix from one of its SPUEs.

**Theorem 2.** Let $(U, \square)$ be a SPUE of $A$. Consider a degree-$d$ complex polynomial $\upsilon(x) = \sum_{n=0}^d a_n T_n(x)$, written in the basis of Chebyshev polynomials. Consider its associated signal processing polynomial $\Upsilon(z) = \sum_{n=0}^d a_n z^n$. Assume that $\max_{z \in \partial B_1(0)} |\Upsilon(z)| \leq 1$. Let $\mathcal{G}$ be the GQSP

(Theorem 1) transformation of the qubitized walk operator of $(U, \square)$ applying the polynomial $\Upsilon$. Then, $(\mathcal{G}, |0\rangle \otimes \square)$ is a PUE of $\upsilon(A)$.

Performances of the previous construction depend on the scaling factor of the target polynomial, as defined in Definition 1.

**Definition 1.** Let $\upsilon$ be a complex-coefficient polynomial and $\Upsilon$ its signal processing polynomial, as introduced in Theorem 2. Define the scaling factor $\beta$ of $\upsilon$ by:

$$\beta = \frac{\max_{z \in \partial B_1(0)} |\Upsilon(z)|}{\max_{x \in [-1, 1]} |\upsilon(x)|}. \tag{13}$$

Finally, Proposition 4 defines polynomials of non-symmetric matrices and provides a construction of them using their hermitianizations 4.

**Proposition 4.** Let $(U, \square_L, \square_R)$ be a PUE of $A$. Define $\overline{U} = (|0\rangle\langle0| \otimes U + |1\rangle\langle1| \otimes U^\dagger)(X \otimes 1)$ and $\overline{\square} = |0\rangle\langle0| \otimes \square_L + |1\rangle\langle1| \otimes \square_R$. Then, $(\overline{U}, \overline{\square})$ is a SPUE of $|0\rangle\langle1| \otimes A + |1\rangle\langle0| \otimes A^\dagger$. In particular, $\overline{U}$ can be constructed from a controlled-$U$ gate, its adjoint, and an $X$ gate. $(\overline{U}, \overline{\square})$ is called the hermitianization of $(U, \square_L, \square_R)$.

**Proposition 5.** Let $(U, \square_L, \square_R)$ be a PUE of an operator $A$. Let $(\overline{U}, \overline{\square})$ be the hermitianization of $(U, \square_L, \square_R)$ and consider a degree $d$ complex polynomial $\upsilon$ with even part $\upsilon_e$ and odd part $\upsilon_o$. Let $\mathfrak{G}$ be the GQET applying the polynomial $\upsilon$ to $(\overline{U}, \overline{\square})$. Then,

$$\langle0| \otimes \overline{\square}^\dagger \mathfrak{G} |0\rangle \otimes \overline{\square} = \begin{pmatrix} W^\dagger \upsilon_e(\Sigma)W & W^\dagger \upsilon_o(\Sigma)V \\ V^\dagger \upsilon_o(\Sigma)^\dagger W & V^\dagger \upsilon_e(\Sigma)V \end{pmatrix}, \tag{14}$$

where $A = W^\dagger \Sigma V$ is the singular value decomposition of $A$.

### Quadratic speedup polynomial
Let us now state formally certain properties of the eigenvalues and singular values of the curved discriminant $\mathcal{D}$ introduced in Paragraph B. The proof of Proposition 6 is simple and given in the Supplementary Information.

**Proposition 6.** Let $\mathcal{D}$ be the discriminant of an ergodic Markov kernel $P$. Then, $\mathcal{D}$ and $P$ are similar. In particular, they share the same spectrum. Also, $\mathcal{D}^\dagger\mathcal{D}$ is similar to $P^\star P$.

The quantum algorithm described in Paragraph B applies a real polynomial $\upsilon_{\epsilon,d}(x) = \epsilon T_d(x T_{1/d}(1/\epsilon))$ to the singular values of $\mathcal{D}$, where $\epsilon \in ]0,1]$ [and for $y \in ]0, 1]$, $T_y : [1, \infty[ \to \mathbb{R}$ is $x \mapsto \cosh\left(y \cosh^{-1}(x)\right)$. The polynomial is such that $\max_{x \in [-1,1]} |\upsilon_{\epsilon,d}(x)| = 1$, $\max_{x \in [-1+\delta, 1-\delta]} |\upsilon_{\epsilon,d}(x)| = \epsilon$, $|\upsilon_{\epsilon,d}(\pm 1)| = 1$ and of degree $d \in \mathcal{O}\left(\delta^{-1/2} \log(1/\epsilon)\right)$. Applying such a polynomial using the GQSVT formalism requires the signal processing polynomial $\Upsilon_{\epsilon,d}$ to verify the scaling condition $\|\Upsilon_{\epsilon,d}(e^{i\theta})\| \leq 1$ for all $\theta \in \mathbb{R}$. If this condition is not met, $\upsilon_{\epsilon,d}$ must be divided by some constant, implying a loss in the success probability of the algorithm. The optimal constant is called the scaling factor $\beta_{\epsilon,d}$ and defined by:

$$\beta_{\epsilon,d} = \frac{\max_{z \in \partial B_1(0)} |\Upsilon_{\epsilon,d}(z)|}{\max_{x \in [-1,1]} |\upsilon_{\epsilon,d}(x)|}. \tag{15}$$

In the Supplementary Information, we show the following Corollary 1 stating that $\beta_{\epsilon,d} = 1$.

**Corollary 1.** The polynomial $\upsilon_{\epsilon,d}(x) = \epsilon T_d(x T_{1/d}(1/\epsilon))$ has scaling factor $\beta_{\epsilon,d} = 1$ for any $d \geq 1$ and $0 < \epsilon \leq 1$. Moreover, $\max_{x \in [-1,1]} |\upsilon_{\epsilon,d}(x)| = 1$.

Let us now prove Proposition 2. The proof is based on the following theorem, whose proof is given in the Supplementary Information. Theorem 3 is inspired by a result from ref. [41]. A more precise

version of Proposition 2 can be found in the Supplementary Information.

**Theorem 3.** Let $0 < \epsilon < 1$. Let $(\delta_k)_{k \in \mathbb{N}}$ be the sequence of positive real numbers defined for each $k \in \mathbb{N}$ by:

$$\delta_k = \frac{T_{1/k}(1/\epsilon) - 1}{T_{1/k}(1/\epsilon)}. \tag{16}$$

Let $\mathcal{C}(\epsilon, k)$ be the set of defined parity real polynomials $v$ having all their roots in $]-1, 1[$ and such that:

- $v(1) = 1$,
- $\forall x \in [-1 + \delta_k, 1 - \delta_k], |v(x)| \le \epsilon$.

Then, there is no polynomial of degree less than $k$ in $\mathcal{C}(\epsilon, k)$. Moreover, the polynomial $v_{\epsilon,k}(x) = \epsilon T_k(T_{1/k}(1/\epsilon)x)$ is such that $\max v_{\epsilon,k}(\epsilon) = 1 - \delta_k$ and the only polynomial of degree $k$ in $\mathcal{C}(\epsilon, k)$.

*Proof of Proposition 2.* Consider the sequence of polynomials $(v_{1/4,k})_{k \in \mathbb{N}}$. For each $k \in \mathbb{N}$, let $y_k = \max v_{1/4,k}^{-1}(3/4)$. Let $m(k)$ be the smallest integer $m$ such that $y_k \le 1 - \delta_m$. Note that $v_{1/4,k} \in \mathcal{C}(3/4, m(k) - 1)$ so that $k \ge m(k) - 1$ by Theorem 3. Moreover, the estimate $k \in \Theta\left(1/\sqrt{\delta_k}\right)$ (see the Supplementary Information) implies the existence of constants $c_1, c_2 > 0$ such that $k \ge \frac{c_1}{\sqrt{\delta_k}}$ and $k + 1 \le \frac{c_2}{\sqrt{\delta_k}}$ for $k$ large enough. By definition of $m(k)$,

$$y_k \le 1 - \frac{c_1^2}{m(k)^2} \implies \frac{c_1}{\sqrt{1 - y_k}} \le m(k). \tag{17}$$

Combining the inequalities:

$$\frac{1 - y_k}{\delta_k} \ge \left(\frac{c_1}{c_2}\right)^2 > 0. \tag{18}$$

As shown in ref. 42, there exists a real polynomial $q_\epsilon$ of degree $\mathcal{O}(\log(1/\epsilon))$ such that:

- $|q(x)| \le 1$ for all $x \in [-1, 1]$,
- $|q(x)| \le \epsilon$ for all $x \in [-1, 1/4]$, and
- $q(x) \ge 1 - \epsilon$ for all $x \in [3/4, 1]$.

For $k$ odd, consider the polynomial $r_{k,\epsilon} = q_\epsilon \circ v_k$ of degree $\Theta\left(\delta_k^{-1/2} \log(1/\epsilon)\right)$. Let $c = (c_1/c_2)^2$. As the composition of two polynomials sending $[-1, 1]$ to a subset of $[-1, 1]$, $r_{k,\epsilon}(x) \in [-1, 1]$ for all $x \in [-1, 1]$. If $x \in [-1, 1 - \delta_k]$, then $v_{1/4,k}(x) \le 1/4$ and $|r_{k,\epsilon}(x)| \le \epsilon$. If $x \in [1 - c\delta_k, 1] \subset [y_k, 1]$, then $v_{1/4,k}(x) \ge 3/4$ and $r_{k,\epsilon}(x) \ge 1 - \epsilon$. Therefore, $r_{k,\epsilon}$ has all the claimed properties.

*End of proof.*

Proposition 2 directly implies Proposition 3.

*Proof of Proposition 3.*

First, note that the spectra $\sigma(D)$ and $\sigma(Q)$ of $D$ and $Q$ are related through the equation $\sigma(D) = \langle \mu | D | \mu \rangle \sigma(Q)$. Therefore, the second largest eigenvalue of $D$ can be written $1 - \delta = 1 - \langle \mu | D | \mu \rangle (1 - \gamma(Q))$. In order to use the polynomial of Proposition 2 to separate the two leading eigenvalues of $D$, it is sufficient to have $1 - c\delta \le \langle \mu | D \mu \rangle$. Reformulating the inequality yields the first claim. The second claim follows from the application of a more general corollary of the variational principle given, stated in the Supplementary Information.

*End of proof.*

## Mixed kernels

Note that the flat discriminant is common to a Markov kernel and its time-reversal. Indeed, for each $x, y \in \mathbb{S}$,

$$\sqrt{P(x,y)P(y,x)} = \sqrt{\frac{\pi(y)P^\star(y,x)\pi(x)P^\star(x,y)}{\pi(x)\pi(y)}}$$
$$= \sqrt{P^\star(x,y)P^\star(y,x)}. \tag{19}$$

Assume, for example, that the kernel we are interested in is not mixed, but its time-reversal $P^\star$ is. Then, the reflection through the most reversible distribution can be constructed with a constant number of steps. Moreover, this distribution has a large overlap with $\pi$. This property is especially interesting if $P^\star$ is not explicitly known. Corollary 2 states it more formally.

**Corollary 2.** Let $0 < \epsilon < 1/(2\sqrt{8})$ and $t = \min(\tau(\epsilon), \tau^\star(\epsilon))$ where $\tau(\epsilon)$ is the Hellinger mixing time of $P$ and $\tau^\star(\epsilon)$ is the mixing time of $P^\star$. It is possible to implement the reflection $2|\mu(t)\rangle\langle\mu(t)| - 1$ up to spectral norm error $\eta > 0$ with $\mathcal{O}(\log(1/\eta))$ uses of $R$ the Szegedy quantum walk operator associated with $P^\sharp$. Moreover, $\langle\mu(t)|\pi\rangle \ge 1 - 3\sqrt{8}\epsilon + \mathcal{O}(\epsilon^2)$.

*Proof.* Without loss of generality, assume that $t = \tau(\epsilon)$. Following the reasoning from the proof of Lemma 20 in ref. 43, we get that

$$\| \square - \square_\Pi \| \le \sqrt{2}\epsilon. \tag{20}$$

As a consequence, writing $\Pi$ for the perfectly mixed kernel,

$$\| D - |\pi\rangle\langle\pi| \| = \| \square^\dagger S \square - \square_\Pi^\dagger S \square_\Pi \|$$
$$\le \| \square^\dagger S \square - \square^\dagger S \square_\Pi \| + \| \square^\dagger S \square_\Pi - \square_\Pi^\dagger S \square_\Pi \|$$
$$\le \| \square - \square_\Pi \| + \| \square^\dagger - \square_\Pi^\dagger \|$$
$$\le \sqrt{8}\epsilon. \tag{21}$$

By the Bauer-Fike Theorem (see Supplementary Information), $|\lambda_1| \le \sqrt{8}\epsilon$, and the leading eigenvalue of $D_t$ is well separated from the others for $\epsilon$ small enough. However, $\max_{x \in \mathbb{S}} d_{TV}(P^t(x, \cdot), \pi) \le \sqrt{2}\epsilon$. Then, $\langle\pi|D|\pi\rangle \ge 1 - \sqrt{8}\epsilon$. Note also that $\mu - \lambda_1 \le 1 + \sqrt{8}\epsilon$. As detailed in the Supplementary Information, we arrive at:

$$\langle\pi|\mu(t)\rangle^2 \ge \frac{1 - 2\sqrt{8}\epsilon}{1 + \sqrt{8}\epsilon} = 1 - 3\sqrt{8}\epsilon + \mathcal{O}(\epsilon^2). \tag{22}$$

*End of proof.*

## Random walks on groups

The presented quantum algorithms have a complexity that is governed by spectral properties of $D$. Being a symmetric matrix, those eigenvalues are necessarily in $[-1, 1]$, whereas the nontrivial eigenvalues of an arbitrary Markov kernel can lie anywhere in $B_1(0)$, the unit disk. As such, the eigenvalues of a kernel and its flat discriminant may be quite different. This paragraph introduces random walks on groups and shows that in the presence of symmetries, the eigenvalues of the flat discriminant are in fact very close to those of the additive reversibilization of the chain.

**Definition 2.** Consider a finite group $(\mathbb{S}, \circ)$ and a probability distribution $v$ on $\mathbb{S}$. Let $(Z_t)_{t \ge 1}$ be independent and identically distributed random variables with law $v$, and consider the process $X = (X_t)_{t \ge 1}$ with initial condition $X_0 \in \mathbb{S}$ defined by:

$$X_t = Z_t \circ Z_{t-1} \circ \dots \circ Z_1 \circ X_0. \tag{23}$$

Then $X$ is a Markov chain on $\mathbb{S}$ called the random walk on $(\mathbb{S}, \circ)$ with increment law $\nu$. Its transition kernel is given for all $x, y \in \mathbb{S}$ by $P(x, y) = \nu(y \circ x^{-1})$.

We prove the following result in the Supplementary Information.

**Proposition 7.** Assume that $P$ is a random walk on a group $(\mathbb{S}, \circ)$. Then, $\langle \pi | \mu \rangle = 1$ and $\| D - \mathcal{D}_A \| = 1 - \langle \pi | D | \pi \rangle$, where $D$ is the flat discriminant of $P$ and $\mathcal{D}_A$ is the curved discriminant of $P_A = (P + P^\star)/2$. In particular,

$$\sigma(D) \subset \bigcup_{\lambda \in \sigma(P_A)} B_{1-\langle \pi|D|\pi\rangle}(\lambda). \tag{24}$$

Also, if $I \subset \sigma(P_A)$ and

$$\left( \bigcup_{\lambda \in I} B_{1-\langle \pi|D|\pi\rangle}(\lambda) \right) \bigcap \left( \bigcup_{\lambda \in \sigma(P_A) \setminus I} B_{1-\langle \pi|D|\pi\rangle}(\lambda) \right) = \emptyset, \tag{25}$$

then there are exactly $|I|$ eigenvalues of $D$ in $\bigcup_{\lambda \in I} B_{1-\langle \pi|D|\pi\rangle}(\lambda)$.

### Quasi-stationary distributions

In practice, Markov kernels often present a long mixing time because of the presence of regions of the state space that are hard to escape. Within such a region, the probability measure of the process conditioned on the fact that it did not exit may however, converge rapidly to some locally supported distribution. Definition 3 defines such distributions properly and Proposition 8 shows that their existence is favorable for $\langle \pi | D_j | \pi \rangle$ to be close to 1 for $j$ much smaller than the mixing time.

**Definition 3.** Let $(X_t)_{t \in \mathbb{N}}$ be a Markov chain with state space $E \cup \partial$ (with $\partial \cap E = \emptyset$) which is absorbed at $\partial$ (i.e., $\mathbb{P}_\partial(X_1 = \partial) = 1$). A quasi-stationary distribution is a probability measure $\nu$ on $E$ such that:

$$\forall t \in \mathbb{N}, \forall A \subset E : \nu(X_t \in A | t < \tau_\partial) = \nu(A), \tag{26}$$

where $\tau_\partial = \inf\{t \geq 0, X_t \in \partial\}$ is the absorption time of $X$.

Write the state space as $\mathbb{S} = \cup_{i=1}^m E_i \cup \partial$ for some $m \in \mathbb{N}$ and assume the Markov kernel has quasi-stationary distributions in each of the $E_i$ (with complementary set $\partial_i = \cup_{j=1, j \neq i}^m E_j \cup \partial$, absorption time $\tau_{\partial_i}$ and stationary distribution $\nu_i$). $E = \cup_{i=1}^m E_i$ is a disjoint union.

**Proposition 8.** The following inequality holds for any integer $j \geq 1$:

$$\langle \pi | D_j | \pi \rangle \geq \pi(E) \min_{1 \leq i \leq m} \min_{x \in E_i} \mathbb{P}_x(j < \tau_{\partial_i})$$

$$\min_{1 \leq i \leq m} \left( 1 - 2 \mathbb{E}_{\pi_{|E_i}} [d_{TV}(\mathbb{P}_x(X_j = \cdot | j < \tau_{\partial_i}), \nu_i)] \right. \tag{27}$$

$$\left. - d_{TV} \left( \pi_{|E_i} \otimes \nu_i, \nu_i \otimes \pi_{|E_i} \right) \right).$$

In the Supplementary Information, we prove Proposition 8. We also prove an often tighter result. However, the probabilistic interpretation of this other result is not as clear because the total variation distance is replaced by another distance.

The present study is supplemented by a mathematically comprehensive Supplementary Information. The Supplementary Information contains a proof that the scaling factor of the method using $R$ and $R^\star$ is precisely 1, gives a tighter construction for Proposition 3, and locates the eigenvalues of the flat discriminant in different contexts. The theoretical results are illustrated with analytic and numerical experiments.

### Data availability

Data generated during the study is available upon request from the authors (E-mails: baptiste.claudon@qubit-pharmaceuticals.com or jean-philip.piquemal@sorbonne-universite.fr).

### Code availability

The code used during the study is available upon request from the authors (E-mails: baptiste.claudon@qubit-pharmaceuticals.com or jean-philip.piquemal@sorbonne-universite.fr).

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

## Acknowledgements

This work has been funded in part by the European Research Council (ERC) under the European Union's Horizon 2020 research and innovation program (Grant No. 810367), project EMC2 (J.P.P.). Support from the PEPR EPIQ—Quantum Software (ANR-22-PETQ-0007, J.P.P.) and HQI (J.P.P.) programs is also acknowledged.

## Author contributions

B.C., J.P.P., and P.M. conceived and designed the experiments. B.C. performed the numerical experiments. B.C. analyzed the data. B.C. wrote the paper with the inputs of J.P.P. and P.M. J.P.P. and P.M. supervised the work.

## Competing interests

J.P.P. is a shareholder and co-founder of Qubit Pharmaceuticals. The remaining authors declare no other competing interests.
