## [Transparent Peer Review file · Nature Communications]

Quantum Speedup for Nonreversible Markov Chains

Corresponding Author: Professor Jean-Philip PIQUEMAL

Version 0:

Reviewer comments:

Reviewer #1

(Remarks to the Author)

- What are the noteworthy results?

The authors propose two quantum algorithms for sampling the stationary distribution of a Markov chain. When the Markov chain is irreversible with respect to the target stationary distribution, the second algorithm can provide an up-to-exponential speedup and does not require knowing the stationary distribution up to a normalization constant. The resulting algorithmic acceleration goes beyond the quadratic speedup known for reversible chains but hinges on the reversibility on π -average condition.

- Will the work be of significance to the field and related fields? How does it compare to the established literature? If the work is not original, please provide relevant references.

Yes, the work addresses the efficient sampling of stationary distributions of generic Markovian systems, including those possibly breaking the detailed balance, which yields important implications in various domains of physical and information sciences. The first result summarized in Proposition 1 appears related to previous results on Grover-type quadratic speedups. It would be interesting to compare and distinguish from the established literature.

- Does the work support the conclusions and claims, or is additional evidence needed?

Analytical and numerical examples were demonstrated to showcase the more-than-quadratic speedup of the sampling algorithm. However, the work does not seem to fully support, on a broader scope, the claim of impact across various fields as it remains unclear for which applications the assumptions made in the analysis are valid.

- Are there any flaws in the data analysis, interpretation and conclusions? - Do these prohibit publication or require revision?

The paper reads a bit dense and heavy of mathematical notations. It would be helpful to (i) restructure the paper to highlight the main result, (ii) include an overview figure that schematically illustrates the core algorithm, and (iii) summarize and clearly distinguish the novel contributions in the paper.

It remains a bit unclear what are some sufficient and necessary conditions for reversibility on π -average, since this is a key condition for the efficiency of the sampling. Related to this, the method is illustrated only for a few contrived toy examples and the claim of broader applicability to statistics, machine learning, and computational modeling is not immediately obvious.

Minor typos need to be corrected throughout the main text E.g.,

$\partial B_1(0)$ as the complex unit circle in the right column of page 2 is undefined.

$1-|P(z)|^2$ in the paragraph discussing the complementarity condition on page 4 needs to be revised, as P is now defined as the Markov transition kernel.

- Is the methodology sound? Does the work meet the expected standards in your field?

The mathematical (Markov chain theory) and algorithmic (projected unitary encoding and generalized quantum eigenvalue transformation) tools used for implementing the sampling algorithm are well established and developed.

- Is there enough detail provided in the methods for the work to be reproduced?

The work includes extensive mathematical proofs but does not provide source code. However, it offers comprehensive details presented mostly in a self-consistent manner, likely enabling the exact reproduction of the algorithm and numerical results.

Reviewer #2

(Remarks to the Author)

Reviewer #3

(Remarks to the Author)

This paper introduces quantum algorithms that provide up-to-exponential speedup for sampling from the stationary distribution of nonreversible Markov chains. It extends existing method to non-reversible chains.

The core algorithmic components are the Generalized Quantum Eigenvalue Transform (GQET) and the Generalized Quantum Singular Value Transform (GQSVT), which facilitate the construction of reflections through the stationary distribution.

Quantizing nonreversible Markov chains is an important problem since nonreversible chains can be more efficient and may exhibit faster mixing. While I appreciate the effort put into this work, I have several concerns about its technical aspects.

1) I would like to see a stronger motivation for the second method, which addresses cases where the stationary distribution is unknown up to a normalizing constant. From my experience, most papers on Markov chain algorithms in physics, machine learning, and statistics assume that the stationary distribution is known up to a normalizing constant, as is standard in statistical physics, Bayesian learning, and many relevant models. This makes me question why so much effort is dedicated to the second case.

2) The technical section is difficult to follow, and the novelty appears somewhat limited. From my reading, it seems (at least for the first method) that the authors simply apply the GQSVT algorithm, and the results follow directly. Therefore I have three suggestions

2.1) The authors may consider providing a more comprehensive overview of existing work, particularly on GQSVT and projected unitary encoding, as these appear to be central to their approach. Currently, the technical background section is too terse. In fact, while I am very familiar with about 70% of the material, the remaining 30% is impossible to follow. (see also comment 3)

2.2) The authors should clarify how they utilize existing algorithms and whether their approach is merely a direct application of these methods.

2.3) The authors could improve clarity by helping readers "connect the dots." Even after reviewing the original GQSVT paper, it is not entirely clear to me why Proposition 1 holds. For instance, I assume the authors apply GQSVT to the matrix $D = \sqrt{PP^*}$, but it is unclear why this corresponds to a reflection over π and how it approximates π .

3) I may be overlooking something, but I am unable to verify some of the claims made in the paper. For example, at the end of the second paragraph in Section 2, the statement "... Indeed, ..., .. a comprehensive exposition" does not seem to hold in a straightforward manner. If I take the naive encoding with $\square = I$ and $U = A$, the claim suggests that (A^2, I) is a projected unitary encoding of $2A - I$. However, since $A^2 = A$, this does not match $2A - I$?

4) I wonder whether GQSVT is necessary? For example, can we directly use the easier operator in e.g., Section 17.2 of <https://www.cs.umd.edu/~amchilds/qa/qa.pdf> ? It seems this operator does not explicitly require the chain to be reversible.

Reviewer #4

(Remarks to the Author)

This work generalizes the quantum speedup for Markov chain simulations, by proposing a way to apply it not only to reversible Markov chains but also to non-reversible ones. This aim and the associated important applications are quite clear.

The results themselves though are not so clear to follow.

A first result, it is said, needs to know the stationary distribution (i.e. the one that we are trying to sample from). Once this is known, there should be much more efficient ways to sample from it than MCMC! E.g. without taking any walk constraints. This is not discussed.

A second result does not need such knowledge. Still, the formulation of the propositions (for instance, starting by postulating the Markov kernel P) does not make it clear which knowledge of P is required in order to select the polynomial and other parameters.

In particular, when making statements like " P and P^* have roughly the same high maxima", it is easy to imagine counterexamples to this. This is probably why conditions are needed for the scheme to work. But how would one know if the conditions are satisfied or not, on a given problem instance? if/when we can trust the results of such scheme?

In this sense, the technical properties highlighted in this paper contain interesting points, but their relevance in a full

application context is not explained as clearly as it should be. I would advise the authors to revise the writing style in Section 2, putting emphasis on what must be known or not and how one would use the construction in practice (facing a precise oracle). Maybe, illustrate on a running example how all things work what is needed or not. Especially for this type of journal, this sort of intermediate yet precise level, between introduction and technical proofs, is the most important part for the readers and it is rather missing here.

Version 1:

Reviewer comments:

Reviewer #1

(Remarks to the Author)

I thank the authors for their efforts in revising the manuscript, "Quantum Speedup for Nonreversible Markov Chains." They have successfully addressed several key issues, particularly in clarifying the paper's motivations and contributions. While the exposition of the technical background is significantly improved, it would benefit from some further revisions. My specific recommendations are listed below.

Figure 1

This figure helps make the paper's ideas more accessible. However, it could be further improved by:

1. Adding labels that connect to the main text, such as referencing Proposition 1 and Proposition 3 in the corresponding columns.
2. Using consistent terminology from the main text, such as "curved/flat discriminant," "(symmetric) projected unitary encoding," and "projector."
3. Defining terms used in the figure, such as "fast-forwarding polynomial" and "leading eigenvalue selection polynomial," within the main text.
4. Reorganizing the layout. The plots in the middle are positioned in a way that makes it visually unclear which algorithm they relate to.
5. Explicitly adding a box representing the GQSVT/GQET step to the flowchart.

Figures 2 & 4

Figure 2(b): Adding a plot of $\gamma(Q_j)$ and $1 - \langle \pi | D_j | \pi \rangle$ (instead of $\langle \pi | D_j | \pi \rangle$) would help confirm the condition indicated by the light gray line.

Figure 4: The figure would be more readable if it plotted the directly relevant quantities, namely $1 - \langle \pi | D_j | \pi \rangle$ and $\gamma(Q_j)$, rather than the related quantities $1 - \gamma(Q_j)$ and $\langle \pi | D_j | \pi \rangle$. Furthermore, the figure does not appear to support the statement, "the overlap between the most reversible distribution $\mu(j)$ and the target state $|\pi\rangle$ is large from the moment this condition is verified," as the overlap (the dashed blue line) appears to be large for all values of j .

Clarity and Terminology

As Referee 3 noted, adding pointers throughout the text to help readers connect concepts would be beneficial. While the revised version is a significant improvement, some sections could still benefit from further clarification and more consistent introduction of terminology before its use. For example:

* Q is formally defined in Eq. (3), but the term "geometric reversibilization" is not introduced there.

* The terms "fast-forwarding polynomial" and "leading eigenvalue selection polynomial" should be properly defined in the text.

* The definition for "additive reversibilization" currently appears only in the caption of Figure 2. I recommend moving this definition into the main text, possibly to Section 2.a.

Section-Specific Comments

Section 4: Consider renaming this section to "Eigenvalue transformations of the flat discriminant" to better align with the GQET formalism used.

Section 5: This section ("Sequences of flat discriminants") remains difficult to follow and would benefit most from another revision.

* The motivating example of the Markov chain on the N -point circle is unclear. Assuming a Markov chain on a 3-point circle such that $P = \frac{1}{2} \begin{pmatrix} 1 & 1 & 0 \\ 0 & 1 & 1 \\ 1 & 0 & 1 \end{pmatrix}$, then $P^{P^T} = \frac{1}{4} \begin{pmatrix} 2 & 1 & 1 \\ 1 & 2 & 1 \\ 1 & 1 & 2 \end{pmatrix}$. I fail to see how $D = \sqrt{P^{P^T}}$ would be $\frac{1}{2}I$.

There may be an elementary misunderstanding on my part that merits clarification.

* I also do not follow the motivation for studying the sequence D_j associated with P^j ; additional pointers seem necessary here.

Typo

* On page 5, there is a stray "6" immediately following "Perron-Frobenius."

Reviewer #3

(Remarks to the Author)

Thanks for the revision. The revised version has addressed most of my concerns.

Reviewer #4

(Remarks to the Author)

While the revision improves the manuscript in some respects, I believe it still falls short of the standard expected for publication in Nature Communications.

I agree with the other reviewers that the paper builds on existing quantum routines, but I also recognize that it raises an interesting point: how a suitable framework can be constructed to apply these routines effectively and achieve a speedup. In this regard, the main contribution—currently maybe underemphasized—might be better framed as a structural or methodological insight into how properties of a nonreversible Markov chain P can be deduced from associated reversible constructs.

More importantly though, a few specific issues remain unclear or insufficiently addressed:

1. Positioning of the Work within the Broader Landscape:

The introduction still too quickly defaults to the statement that MCMC is the paradigmatic method for sampling from complex stationary distributions. This framing misses an opportunity to better contextualize the contribution within the broader landscape of both classical and quantum methods for stationary distribution estimation.

In particular, there exist quantum accelerations which directly sample from the stationary state, without relying on speeding up a long time evolution. There are also classical techniques—especially for nonreversible chains—that exploit structural modifications to achieve speedups. Without a clear comparison to these alternatives, the risk is that the proposed quantum "speedup" is only relative to a limited or overly simplistic benchmark.

2. The analysis of P^{*} and its quantum implementation is interesting, but there are two points that require clarification:

(a) The paper acknowledges in the introduction that nonreversible Markov chains can (often) mix faster than reversible ones, yet the proposed speedup is framed through reversibilization, using P^{*} . Furthermore, this part is motivated by stating a bound on the mixing of nonreversible Markov chains which is significantly worse in terms of the spectral gap, than the reversible case. Highlighting this worst-case scenario seems at odds with the earlier motivation --- and with approaches which indeed use nonreversibility (+other gadgets, like selecting a random stopping time) to speed up mixing.

To avoid confusion, it would be helpful to more clearly define the scope of the results: for which classes of Markov chains is the proposed approach most relevant? Can the authors specify when reversibilization offers an advantage, and when it may obscure potential benefits of nonreversible dynamics? Without this clarification, the paper risks appearing inconsistent in its arguments, depending on the context being considered.

(b) Regarding the actual implementation of such algorithm: I may not have been sufficiently clear in my earlier comments. The key question is about the oracle access model: how is $P(x,y)$ made available to the quantum algorithm? For the approach to be efficient and practically meaningful, this cannot involve listing all components of the transition matrix. Once a specific oracle model is fixed (e.g., querying $P(x,\cdot)$ for a given x), the authors should clarify how this enables implementation of the quantum walk based on P^{*} , and what assumptions are needed to compute or approximate the necessary components.

3. Overlap Argument in Geometric Reversibilization:

The discussion surrounding the expected overlap between μ and π in the geometric reversibilization section appears to rely on an assumption that may not hold in general. The intuition that both P and P^{*} concentrate near maxima of the stationary distribution more rapidly than the mixing time is certainly valuable. However, this ensures high overlap in the sense of the computed formula, only if both chains converge towards the SAME local maximum when starting from the SAME position (this same argument is correctly used when explaining how D_j , and in fact D_j until j is of order N , may say nothing about the stationary distribution in the cycle graph example).

This weakens the general applicability of the argument and suggests that the advantage may only materialize in near-

reversible or specifically structured cases. Leaving this point open is too much of a gamble about the relevance of this algorithm.

In summary, the manuscript presents a collection of interesting observations, and the idea of leveraging reversibilization in quantum Markov chain methods is conceptually appealing. However, the presentation currently lacks the clarity and generality required to convincingly establish a substantial advance suitable for Nature Communications. I would encourage the authors to further refine the scope of their claims, clarify key assumptions (particularly around oracle models, and applicability for the geometric reversibilization), and offer a more balanced comparison to existing classical and quantum approaches.

Version 2:

Reviewer comments:

Reviewer #1

(Remarks to the Author)

I thank the authors for their efforts in revising the manuscript and addressing my concerns.

Reviewer #4

(Remarks to the Author)

After reading the reply to referees, I must say that the paper seems now much clearer to me; and hopefully this is representative of more general readers' opinion.

Most arguments are clarified and I appreciate the additions about the context.

There is just one technical point which I found too sketchy, namely about the oracle access. Indeed, the reference provided explains how to use the oracle of a reversible Markov chain, while here the purpose is precisely to treat non-reversible ones. It would seem necessary to explain how, having access to the "P" oracle, one would construct the "P ^star" oracle. Indeed, in general, it is not obvious (to me) that being able to predict where the walk will go next, implies you could as easily deduce where the walk is coming from.

I may just be overlooking some obvious property; or maybe not. If this point was clarified, all my comments would be resolved.

Version 3:

Reviewer comments:

Reviewer #4

(Remarks to the Author)

I just had one remaining concern: how, in general, an oracle model for the direct chain would imply the oracle model for the reverse chain.

With respect to the material presented now by the authors, I was worried about two things:

- how to convert the transition probability, which thus indeed assumes we can evaluate $\pi(x)/\pi(y)$ for neighboring states; why not, in many settings.

- but also: if given x , we can easily find all y for which $P(x,y) \neq 0$, this would not necessarily imply that given y , we can efficiently find all x for which $P(x,y) \neq 0$.

I therefore found it important to clarify that, with respect to the most general setting, we have to assume these things.

The authors now correctly clarify that this is an additional assumption, which often holds in practice. With this I see no further issues.

Referee 1

Comment: - *What are the noteworthy results?*

The authors propose two quantum algorithms for sampling the stationary distribution of a Markov chain. When the Markov chain is irreversible with respect to the target stationary distribution, the second algorithm can provide an up-to-exponential speedup and does not require knowing the stationary distribution up to a normalization constant. The resulting algorithmic acceleration goes beyond the quadratic speedup known for reversible chains but hinges on the reversibility on π -average condition.

Comment: - *Will the work be of significance to the field and related fields? How does it compare to the established literature? If the work is not original, please provide relevant references.*

Yes, the work addresses the efficient sampling of stationary distributions of generic Markovian systems, including those possibly breaking the detailed balance, which yields important implications in various domains of physical and information sciences. The first result summarized in Proposition 1 appears related to previous results on Grover-type quadratic speedups. It would be interesting to compare and distinguish from the established literature.

Response: We thank the reviewer for the suggestion. We have significantly revised the paragraph surrounding Proposition 1 to clarify its motivation and contribution. In particular, we now emphasize that performance analysis in the nonreversible (and more generally nonnormal) case can only be made rigorous through a newly introduced notion of spectral gap adapted to this broader setting. This allows us to quantify algorithmic performance even when the relationship between the eigenvalues of the kernel and that of the qubitized walk operator is not as simple as in the reversible case.

Changes in the manuscript:

- Paragraph 3 entirely rewritten.
-

Comment: - *Does the work support the conclusions and claims, or is additional evidence needed?*

Analytical and numerical examples were demonstrated to showcase the more-than-quadratic speedup of the sampling algorithm. However, the work does not seem to fully support, on a broader scope, the claim of impact across various fields as it remains unclear for which applications the assumptions made in the analysis are valid.

Response: We agree that full theoretical guarantees for end-to-end application scenarios are difficult to establish. This limitation, however, is not unique to our work and is a common challenge when applying Markov chain methods in practice. To address this, we have expanded the discussion section to better highlight open questions and potential avenues for future work, especially concerning the practical validity of our assumptions in real-world applications.

Changes in the manuscript:

- Added section 6.b. entitled "Research directions".
-

Comment: - *Are there any flaws in the data analysis, interpretation and conclusions? - Do these prohibit publication or require revision?*

The paper reads a bit dense and heavy of mathematical notations. It would be helpful to (i) restructure the paper to highlight the main result, (ii) include an overview figure that schematically illustrates the core algorithm, and (iii) summarize and clearly distinguish the novel contributions in the paper. It remains a bit unclear what are some sufficient and necessary conditions for reversibility on π -average, since this is a key condition for the efficiency of the sampling. Related to this, the method is illustrated only for a few contrived toy examples and the claim of broader applicability to statistics, machine learning, and computational modeling is not immediately obvious. Minor typos need to be corrected throughout the main text E.g., $\partial B_1(0)$ as the complex unit circle in

the right column of page 2 is undefined. 1— $P(z)$ —2 in the paragraph discussing the complementarity condition on page 4 needs to be revised, as P is now defined as the Markov transition kernel.

Response: In response to this comment, we have reorganized the manuscript to improve clarity and accessibility. (i) We have compiled all relevant material from the existing literature into an expanded technical background section. Sections 3, 4, and 5 contain the core novel results and are now structured to directly address the central research question posed in the introduction. (ii) We have included a new figure to illustrate the algorithmic flows of the two approaches. (iii) A new concluding section explicitly summarizes the novel contributions and places them in the broader context of existing work.

Changes in the manuscript:

- Figure to illustrate the algorithmic workflow.
- Revised Conclusion section.

Comment: - *Is the methodology sound? Does the work meet the expected standards in your field? The mathematical (Markov chain theory) and algorithmic (projected unitary encoding and generalized quantum eigenvalue transformation) tools used for implementing the sampling algorithm are well established and developed.*

Comment: - *Is there enough detail provided in the methods for the work to be reproduced? The work includes extensive mathematical proofs but does not provide source code. However, it offers comprehensive details presented mostly in a self-consistent manner, likely enabling the exact reproduction of the algorithm and numerical results.*

Referee 2

Comment: *I co-reviewed this manuscript with one of the reviewers who provided the listed reports. This is part of the Nature Communications initiative to facilitate training in peer review and to provide appropriate recognition for Early Career Researchers who co-review manuscripts.*

Response: We thank the reviewer for evaluating this work.

Referee 3

Comment: *This paper introduces quantum algorithms that provide up-to-exponential speedup for sampling from the stationary distribution of nonreversible Markov chains. It extends existing method to non-reversible chains.*

The core algorithmic components are the Generalized Quantum Eigenvalue Transform (GQET) and the Generalized Quantum Singular Value Transform (GQSVT), which facilitate the construction of reflections through the stationary distribution.

Quantizing nonreversible Markov chains is an important problem since nonreversible chains can be more efficient and may exhibit faster mixing. While I appreciate the effort put into this work, I have several concerns about its technical aspects.

Comment: *1) I would like to see a stronger motivation for the second method, which addresses cases where the stationary distribution is unknown up to a normalizing constant. From my experience, most papers on Markov chain algorithms in physics, machine learning, and statistics assume that the stationary distribution is known up to a normalizing constant, as is standard in statistical physics, Bayesian learning, and many relevant models. This makes me question why so much effort is dedicated to the second case.*

Response: To strengthen the motivation for the second method, which does not assume knowledge of the stationary distribution up to normalization, we have added examples from biological systems and financial modeling where such knowledge is typically unavailable. Moreover, we discuss its relevance in out-of-equilibrium statistical physics, where the stationary distribution is often complex or analytically inaccessible, yet one is still interested in efficient sampling.

Changes in the manuscript:

- 6. Discussion b. Research direction.
-

Comment: *2) The technical section is difficult to follow, and the novelty appears somewhat limited. From my reading, it seems (at least for the first method) that the authors simply apply the GQSVT algorithm, and the results follow directly. Therefore I have three suggestions*

Response: We have revised the structure of the paper to center it around a guiding question posed in the introduction. This makes the logical progression of our results more transparent and improves the overall coherence of the presentation.

Changes in the manuscript:

- In the introduction: *Can quantum algorithms accelerate the mixing of nonreversible Markov processes? In summary, we answer this question by:*
 - *analyzing known quantum singular value transforms in the context of nonreversible Markov chains (which requires simulating the time-reversal), and*
 - *introducing quantum eigenvalue transforms to retrieve an approximation of the stationary distribution without simulating the time-reversal.*
-

Comment: *2.1) The authors may consider providing a more comprehensive overview of existing work, particularly on GQSVT and projected unitary encoding, as these appear to be central to their approach. Currently, the technical background section is too terse. In fact, while I am very familiar with about 70% of the material, the remaining 30% is impossible to follow. (see also comment 3)*

Response: In response to the reviewer's concern, we have significantly expanded and clarified the technical background section. The updated version now provides a more comprehensive overview of

the tools used, including Generalized Quantum Singular Value Transformation (GQSVT) and projected unitary encodings, ensuring that readers unfamiliar with all aspects of these techniques can still follow the technical development.

Changes in the manuscript:

- Clarified technical background section.

Comment: 2.2) *The authors should clarify how they utilize existing algorithms and whether their approach is merely a direct application of these methods.*

Response: We have now clarified how our approach builds upon and extends existing quantum algorithmic techniques. On the classical side, we study the spectral properties of the flat discriminant matrix, while on the quantum side, we construct specific polynomials to enable spectral gap amplification in a manner reminiscent of Grover's algorithm. These clarifications are integrated into the revised sections and proofs.

Comment: 2.3) *The authors could improve clarity by helping readers "connect the dots." Even after reviewing the original GQSVT paper, it is not entirely clear to me why Proposition 1 holds. For instance, I assume the authors apply GQSVT to the matrix $D = \sqrt{PP^*}$, but it is unclear why this corresponds to a reflection over π and how it approximates π .*

Response: To improve readability and help readers "connect the dots," we have added a new explanatory paragraph before Proposition 1. This paragraph clarifies the use of GQSVT, the construction of the reflection operator, and how this enables the preparation of the stationary distribution. We have also moved the detailed proofs of Propositions 2 and 3 to the Methods section for better structural flow.

Changes in the manuscript:

- New paragraph before Proposition 1.
- Proofs of Propositions 2 and 3 moved to the Methods section.

Comment: 3) *I may be overlooking something, but I am unable to verify some of the claims made in the paper. For example, at the end of the second paragraph in Section 2, the statement "... Indeed, ..., .. a comprehensive exposition" does not seem to hold in a straightforward manner. If I take the naive encoding with $\square = I$ and $U = A$, the claim suggests that (A^2, I) is a projected unitary encoding of $2A - I$. However, since $A^2 = A$, this does not match $2A - I$?*

Response: We appreciate the detailed reading of our assumptions. We have clarified in the manuscript that the projected unitary encoding (PUE) must in fact be a symmetric PUE (SPUE). In the example given, the operator A is assumed to be both unitary and symmetric, which ensures $A^2 = 1$. This corresponds to a reflection through the entire space. For the case where A is a projector (e.g., $A = |\phi\rangle\langle\phi|$), we now provide an explicit computation.

Changes in the manuscript:

- Let us explain why it is sufficient to construct SPUE of $|\pi\rangle\langle\pi|$. Assume that (U, \square) is a SPUE of $A = |\phi\rangle\langle\phi|$, for a normalized state $|\phi\rangle$. Recall the definition of the qubitized walk operator $\mathcal{W} = (2\square\square^\dagger - 1)U$. Compute:

$$\begin{aligned}
 \square^\dagger \mathcal{W}^2 \square &= \square^\dagger U (2\square\square^\dagger - 1) U \square \\
 &= 2\square^\dagger U \square \square^\dagger U \square - \square^\dagger U^2 \square \\
 &= 2|\phi\rangle\langle\phi| \langle\phi|\phi\rangle \langle\phi| - \square^\dagger \square \\
 &= 2|\phi\rangle\langle\phi| - 1.
 \end{aligned} \tag{1}$$

Since \square is a partial isometry, $\square^\dagger \square$ is the projection on its support. Thus, $\square^\dagger \mathcal{W}^2 \square$ acts as $2|\phi\rangle\langle\phi| - 1$ on the support of \square .

Comment: 4) I wonder whether GQSVT is necessary? For example, can we directly use the easier operator in e.g., Section 17.2 of <https://www.cs.umd.edu/~amchilds/qa/qa.pdf>? It seems this operator does not explicitly require the chain to be reversible.

Response: We thank the reviewer for raising this important point. We now emphasize in the introduction that most prior quantum algorithms for Markov chains, particularly those based on continuous-time quantum walks (CTQWs), assume reversibility. Indeed, CTQWs are well-defined if equation 17.7 is a Schrödinger equation. In particular, the Laplacian should be symmetric. The underlying walk must therefore also be symmetric, namely reversible with respect to the uniform distribution. This structural constraint limits the applicability of such approaches to a relatively narrow subclass of chains. By contrast, our methods are designed to handle more general nonreversible kernels, thereby extending quantum sampling techniques beyond the scope of CTQW-based frameworks.

Referee 4

Comment: *This work generalizes the quantum speedup for Markov chain simulations, by proposing a way to apply it not only to reversible Markov chains but also to non-reversible ones. This aim and the associated important applications are quite clear.*

The results themselves though are not so clear to follow. A first result, it is said, needs to know the stationary distribution (i.e. the one that we are trying to sample from). Once this is known, there should be much more efficient ways to sample from it than MCMC! E.g. without taking any walk constraints. This is not discussed. A second result does not need such knowledge. Still, the formulation of the propositions (for instance, starting by postulating the Markov kernel P) does not make it clear which knowledge of P is required in order to select the polynomial and other parameters. In particular, when making statements like " P and P^ have roughly the same high maxima", it is easy to imagine counterexamples to this. This is probably why conditions are needed for the scheme to work. But how would one know if the conditions are satisfied or not, on a given problem instance? if/when we can trust the results of such scheme?*

Response: We thank the reviewer for emphasizing the importance of clearly stating what is assumed to be known about the stationary distribution. We now specify early in the paper that, unless otherwise noted, we understand "knowing the stationary distribution" to mean knowing it up to a multiplicative constant. MCMC are well suited for this setting.

We have also expanded the introduction to better explain the high-dimensional motivation behind our approach. While rigorous parameter selection is indeed challenging, especially in complex or high-dimensional systems, we emphasize that this challenge is not unique to our work and is commonly addressed heuristically in practical applications.

To support practical implementation, we now include a discussion of how Quantum Phase Estimation (QPE) and analytical techniques can assist in estimating spectral properties of the underlying operators. Additionally, we clarify that in the case of kinetic processes, the adjoint kernel P^* corresponds to the process where initial and final velocities are flipped. This physical interpretation underpins the assumption that P and P^* exhibit similar behavior in such systems. A more detailed explanation and an illustrative example are now included in the Appendix to help ground this intuition.

Changes in the manuscript:

- In the introduction: *The first method requires knowledge of π up to a multiplicative constant (in this work, knowledge of π is always understood to be up to a multiplicative constant).*
- Expanded Introduction.
- Expanded Discussion.

Comment: *In this sense, the technical properties highlighted in this paper contain interesting points, but their relevance in a full application context is not explained as clearly as it should be. I would advise the authors to revise the writing style in Section 2, putting emphasis on what must be known or not and how one would use the construction in practice (facing a precise oracle). Maybe, illustrate on a running example how all things work what is needed or not. Especially for this type of journal, this sort of intermediate yet precise level, between introduction and technical proofs, is the most important part for the readers and it is rather missing here.*

Response: We have revised Section 2 to clearly distinguish the roles of R and R^* , the reflection operators used in our construction. Additionally, we now elaborate on how eigenvalue estimates can be obtained in practice using QPE or classical analytic techniques. To build intuition, we include a running example based on a bottleneck process, which helps illustrate how the method works and what inputs are required from a user-facing oracle perspective.

Changes in the manuscript:

- Figure to illustrate the algorithmic workflow.

Referee 1

Comment: *I thank the authors for their efforts in revising the manuscript, "Quantum Speedup for Nonreversible Markov Chains." They have successfully addressed several key issues, particularly in clarifying the paper's motivations and contributions. While the exposition of the technical background is significantly improved, it would benefit from some further revisions. My specific recommendations are listed below.*

Response: We thank the reviewer for their constructive feedback and valuable recommendations.

Comment: ***Figure 1***

This figure helps make the paper's ideas more accessible. However, it could be further improved by: 1. Adding labels that connect to the main text, such as referencing Proposition 1 and Proposition 3 in the corresponding columns. 2. Using consistent terminology from the main text, such as "curved/flat discriminant," "(symmetric) projected unitary encoding," and "projector." 3. Defining terms used in the figure, such as "fast-forwarding polynomial" and "leading eigenvalue selection polynomial," within the main text. 4. Reorganizing the layout. The plots in the middle are positioned in a way that makes it visually unclear which algorithm they relate to. 5. Explicitly adding a box representing the GQSVT/GQET step to the flowchart.

Response: We revised Figure 1 to address these points. Specifically, we now explicitly refer to Section 3 in the left column and Section 5 in the right column. The terminology in the figure has been aligned with that used in the main text. The concept of fast-forwarding is introduced in the last paragraph of the Technical Background section, and the leading-eigenvalue-selection polynomial is now defined before Proposition 2. The layout has been simplified so that each method occupies a dedicated column, and a clear reference to GQST/GQET has been added to show how the projected unitary encodings of the discriminant connect to those of the projectors. We believe these changes make the similarities and differences between the two methods clearer.

Comment: *Figures 2 & 4*

*Figure 2(b): Adding a plot of $\gamma(Q_j)$ and $1 - \langle \pi | D_j | \pi \rangle$ (instead of $\langle \pi | D_j | \pi \rangle$) would help confirm the condition indicated by the light gray line. **Figure 4:** The figure would be more readable if it plotted the directly relevant quantities, namely $1 - \langle \pi | D_j | \pi \rangle$ and $\gamma(Q_j)$, rather than the related quantities $1 - \gamma(Q_j)$ and $\langle \pi | D_j | \pi \rangle$. Furthermore, the figure does not appear to support the statement, "the overlap between the most reversible distribution $|\mu(j)\rangle$ and the target state $|\pi\rangle$ is large from the moment this condition is verified," as the overlap (the dashed blue line) appears to be large for all values of j .*

Response: We now compare $\gamma(Q_j)$ and $1 - \langle \pi | D_j | \pi \rangle$ in both figures. In Figure 4, we found that the overlap is indeed large for all values of j and, in particular, it remains large when the condition is satisfied. To improve readability, we removed the overlap curve and kept only the three most relevant quantities.

Comment: ***Clarity and Terminology***

*As Referee 3 noted, adding pointers throughout the text to help readers connect concepts would be beneficial. While the revised version is a significant improvement, some sections could still benefit from further clarification and more consistent introduction of terminology before its use. For example: * Q is formally defined in Eq. (3), but the term "geometric reversibilization" is not introduced there. * The terms "fast-forwarding polynomial" and "leading eigenvalue selection polynomial" should be properly defined in the text. * The definition for "additive reversibilization" currently appears only in the caption of Figure 2. I recommend moving this definition into the main text, possibly to Section 2.a.*

Response: We have introduced the term "geometric reversibilization" immediately before its defining equation. As mentioned above, the fast-forwarding concept is now introduced in the last paragraph of the Technical Background section, and the leading-eigenvalue-selection polynomial is defined before Proposition 2. We also moved the definition of additive and multiplicative reversibilizations into Section 2.a.

Comment: ***Section-Specific Comments***

** **Section 4:** Consider renaming this section to "Eigenvalue transformations of the flat discriminant" to better align with the GQET formalism used.*

Response: We thank the reviewer for this suggestion and have adopted the proposed section title.

Comment: ** **Section 5:** This section ("Sequences of flat discriminants") remains difficult to follow and would benefit most from another revision.*

Response: Section 5 has been renamed to "Reversibility on π -average and another notion of reversibilization time" and has been reorganized for clarity. We first describe a strategy analogous to that in the GQSVT method: a number of steps, denoted t_{rev} , is performed classically before applying GQET to the corresponding flat discriminant. We then present the example of a nonreversible walk on a graph with a bottleneck, illustrating how the required number of classical steps may be significantly smaller than the mixing time, leading to an overall speedup. Finally, we explain that t_{rev} tends to remain small in the presence of quasi-stationary distributions, when P and its adjoint P^* mix at different rates, or when considering random walks with group structure.

Comment: ** The motivating example of the Markov chain on the N -point circle is unclear. Assuming a Markov chain on a 3-point circle such that $P = \frac{1}{2} \begin{pmatrix} 1 & 1 & 0 \\ 0 & 1 & 1 \\ 1 & 0 & 1 \end{pmatrix}$, then $PP^T = \frac{1}{4} \begin{pmatrix} 2 & 1 & 1 \\ 1 & 2 & 1 \\ 1 & 1 & 2 \end{pmatrix}$. I fail to see how $D = \sqrt{PP^T}$ would be $\frac{1}{2}I$. There may be an elementary misunderstanding on my part that merits clarification.*

Response: We added the following clarification: "Indeed, for every $x \neq y$, $P(x, y) > 0$ implies $P(y, x) = 0$, and therefore $D(x, y) = \sqrt{P(x, y)P(y, x)} = 0$." The misunderstanding arises from the fact that the square root and multiplication should be performed element-wise. Denoting by \circ the element-wise product of matrices, the 3-point circle example becomes:

$$P \circ P^T = \frac{1}{4} \begin{pmatrix} 1 & 1 & 0 \\ 0 & 1 & 1 \\ 1 & 0 & 1 \end{pmatrix} \circ \begin{pmatrix} 1 & 0 & 1 \\ 1 & 1 & 0 \\ 0 & 1 & 1 \end{pmatrix} = \frac{1}{4} \begin{pmatrix} 1 & 0 & 0 \\ 0 & 1 & 0 \\ 0 & 0 & 1 \end{pmatrix}. \quad (1)$$

The claim then follows by taking the square root.

Comment: ** I also do not follow the motivation for studying the sequence D_j associated with P^j ; additional pointers seem necessary here.*

Response: The motivation behind both the curved and flat discriminant approaches is as follows: we perform a number of steps j (the reversibilization time) classically, and then apply GQSVT or GQET to the corresponding discriminant of P^j . In Section 3, we provide a rigorous justification of this method for the curved discriminant approach. The first paragraph of Section 5 clarifies that, in this context, the reversibilization time should correspond to the first instance at which the reversibility on π -average criterion is met. This leads to more accurate results than applying GQET directly to D_1 .

Comment: ***Typo***

** On page 5, there is a stray "6" immediately following "Perron-Frobenius."*

Response: We thank the reviewer for pointing out this typo. The sentence now reads: "[...], then the Perron-Frobenius Theorem (Theorem 6 in the Supplementary Information)[...]."

Referee 3

Comment: *Thanks for the revision. The revised version has addressed most of my concerns.*

Response: We thank the reviewer for their suggestions.

Referee 4

Comment: *While the revision improves the manuscript in some respects, I believe it still falls short of the standard expected for publication in Nature Communications.*

I agree with the other reviewers that the paper builds on existing quantum routines, but I also recognize that it raises an interesting point: how a suitable framework can be constructed to apply these routines effectively and achieve a speedup. In this regard, the main contribution—currently maybe underemphasized—might be better framed as a structural or methodological insight into how properties of a nonreversible Markov chain P can be deduced from associated reversible constructs.

Response: To emphasize our main contribution more clearly, we added the following paragraph to the introduction: *"More precisely, this work provides two ways to construct an approximate reflection through the target coherent state from the Szegedy quantum walk operators [11] (Section 2d details the oracle access model in more details). Its main contributions are the algorithmic workflows presented on Figure 1. In both methods, we perform classical steps of the walk until a certain criterion is met and the chain is "reversible enough". Then, we perform a suitable quantum singular value or eigenvalue transform, producing the desired reflection with a speedup".* We hope that, along with the revised Figure 1, the new presentation emphasizes better our main contribution.

Comment: *1. Positioning of the Work within the Broader Landscape:*

The introduction still too quickly defaults to the statement that MCMC is the paradigmatic method for sampling from complex stationary distributions. This framing misses an opportunity to better contextualize the contribution within the broader landscape of both classical and quantum methods for stationary distribution estimation.

In particular, there exist quantum accelerations which directly sample from the stationary state, without relying on speeding up a long time evolution. There are also classical techniques—especially for nonreversible chains—that exploit structural modifications to achieve speedups. Without a clear comparison to these alternatives, the risk is that the proposed quantum "speedup" is only relative to a limited or overly simplistic benchmark.

Response: We thank the reviewer for this insightful comment. In the previous version, we stated that MCMC is a standard approach for sampling from complex high-dimensional distributions. We have now refined this statement to clarify the specific setting of interest: we focus on cases where the target distribution is known only up to a multiplicative constant, as is common in statistical physics (e.g., Boltzmann distributions of the form $e^{-\beta U(x)}/Z$, where Z is a partition function that is typically intractable to compute). This refinement better positions our work within the broader landscape of classical and quantum methods for stationary distribution estimation. While there exist both classical techniques and quantum algorithms that directly prepare the stationary state, our contribution addresses the question: *can the standard MCMC paradigm be accelerated using quantum algorithmic techniques in this setting?* In particular, our approach is generic and we agree that we do not compare different methods aimed at specific measures π .

The following paragraph was added to the introduction: *"When studying complex and high-dimensional distributions that are only known up to a normalization constant (as in statistical physics), Markov Chain Monte Carlo (MCMC) is among the most popular methods to provide such classical samples [citation]. If not used solely, it is often part of more elaborate algorithms such as Sequential Monte Carlo [citation] or Annealed Importance Sampling [citation]. The quantum coherent state $|\pi\rangle$ associated with the distribution may be prepared using the Quantum Rejection Sampling (QRS) algorithm [citation] but QRS requires bounds on the normalization constants for efficiency".*

Comment: *2. The analysis of PP^* and its quantum implementation is interesting, but there are two points that require clarification:*

(a) The paper acknowledges in the introduction that nonreversible Markov chains can (often) mix faster than reversible ones, yet the proposed speedup is framed through reversibilization, using PP^ . Furthermore, this part is motivated by stating a bound on the mixing of nonreversible Markov chains which is significantly worse*

in terms of the spectral gap, than the reversible case. Highlighting this worst-case scenario seems at odds with the earlier motivation — and with approaches which indeed use nonreversibility (+other gadgets, like selecting a random stopping time) to speed up mixing.

To avoid confusion, it would be helpful to more clearly define the scope of the results: for which classes of Markov chains is the proposed approach most relevant? Can the authors specify when reversibilization offers an advantage, and when it may obscure potential benefits of nonreversible dynamics? Without this clarification, the paper risks appearing inconsistent in its arguments, depending on the context being considered.

Response: We agree with the reviewer that highlighting only the worst-case bound is not the most informative approach. We have therefore revised the discussion to introduce the notion of a pseudo-spectral gap, which allows for a simpler and more meaningful definition of the reversibilization time. The complexity now naturally reflects the geometric mean of the reversibilization time and the mixing time (see Section 3 and Proposition 1). We also emphasize that both classical and quantum approaches are subject to runtime lower bounds of the order of the graph's diameter. Consequently, Markov chains that already mix in a time comparable to their diameter cannot be meaningfully accelerated, though such cases are rare in practice and are typically efficient enough classically. The following paragraph was added to the Discussion: *"Since both classical and quantum algorithms present complexity lower bounds of the order of the diameter of the underlying Markov kernel, there is no hope to accelerate processes that mix in the time required to cross their underlying graph. Processes which achieve this lower bound mix efficiently. Such processes are hard to design classically and are not representative of practical cases"*.

Comment: (b) Regarding the actual implementation of such algorithm: I may not have been sufficiently clear in my earlier comments. The key question is about the oracle access model: how is $P(x, y)$ made available to the quantum algorithm? For the approach to be efficient and practically meaningful, this cannot involve listing all components of the transition matrix. Once a specific oracle model is fixed (e.g., querying $P(x, \cdot)$ for a given x), the authors should clarify how this enables implementation of the quantum walk based on PP^* , and what assumptions are needed to compute or approximate the necessary components.

Response: To clarify the oracle access model, we added the following sentences and citations to the technical background section: *"The operators R and R^* are typically constructed with arithmetic oracles capable of performing the transformation*

$$|x\rangle \rightarrow \sum_{y \in \mathbb{S}} \sqrt{P(x, y)} |x, y\rangle, \quad (2)$$

for every state $x \in \mathbb{S}$ (see for example [citation]). Such oracles are efficiently implementable whenever the transition probabilities are efficiently computable. As demonstrated in [citations] and exemplified in the Appendix, it is often possible to construct these operators much more efficiently."

Comment: 3. Overlap Argument in Geometric Reversibilization:

The discussion surrounding the expected overlap between μ and π in the geometric reversibilization section appears to rely on an assumption that may not hold in general. The intuition that both P and P^* concentrate near maxima of the stationary distribution more rapidly than the mixing time is certainly valuable. However, this ensures high overlap in the sense of the computed formula, only if both chains converge towards the SAME local maximum when starting from the SAME position (this same argument is correctly used when explaining how D , and in fact D_j until j is of order N , may say nothing about the stationary distribution in the cycle graph example). This weakens the general applicability of the argument and suggests that the advantage may only materialize in near-reversible or specifically structured cases. Leaving this point open is too much of a gamble about the relevance of this algorithm.

Response: We agree that the overlap argument is sensitive to the behavior of the chain around local maxima. However, this property needs to hold only on average with respect to π . Because $1/\pi(x)$ is the return time to x for both P and P^* , we expect the chains to remain near high-probability states for sufficiently long periods, leading to rapid overlap convergence in those regions. For low-probability starting states, P and P^* may diverge for longer times, but their contribution to the expectation value is limited. We updated the paragraph to read: *"Note that $\langle \pi | D_j | \pi \rangle$ is the π -averaged overlap between*

the laws $P^j(x, \cdot)$ and $(P^*)^j(x, \cdot)$ as x follows π : $\langle \pi | D_j | \pi \rangle = \mathbb{E}_\pi [\langle P^j(x, \cdot) | (P^*)^j(x, \cdot) \rangle]$. In practice, a chain with local moves tends to stay for a long time near a local maxima of probability (according to the distribution π), regardless of whether it evolves according to P or P^* . In such regions, the overlap between the laws of P and of P^* tends to 1 very fast (1 minus the overlap is in fact the square of a distance). For $x \in \mathbb{S}$ in a low probability region, $P^j(x, \cdot)$ and $(P^*)^j(x, \cdot)$ can remain different until j equals the mixing time. For example, kinetic processes and their time reversal tend to leave local minima of probabilities in opposite directions. However, $\langle \pi | D_j | \pi \rangle$ is an average under the measure π and this phenomenon does not prevent $\langle \pi | D_j | \pi \rangle$ from being close to 1."

Comment: In summary, the manuscript presents a collection of interesting observations, and the idea of leveraging reversibilization in quantum Markov chain methods is conceptually appealing. However, the presentation currently lacks the clarity and generality required to convincingly establish a substantial advance suitable for Nature Communications. I would encourage the authors to further refine the scope of their claims, clarify key assumptions (particularly around oracle models, and applicability for the geometric reversibilization), and offer a more balanced comparison to existing classical and quantum approaches.

Response: We thank the reviewer for their valuable comments, which have significantly helped us improve the quality and presentation of our work.

We have revised the abstract and introduction to emphasize more clearly that the central contribution of our work is the proposed workflow to accelerate nonreversible processes. A new figure has been added to illustrate this workflow. Furthermore, we now provide a more precise analysis of the performance of the QSVT method, introducing the notion of reversibilization times in a well-defined manner. Section 3, and in particular Proposition 1, has been clarified to better convey the main results. Finally, we offer an intuitive justification for the use of geometric reversibilization by highlighting the similarity between the two approaches. Our workflows are particularly advantageous when the corresponding reversibilization times are smaller than the mixing time of the chain.

We discuss in greater depth the generality of our workflow. In particular, we emphasize that the reversibilization time is always upper-bounded by the mixing time, so there is no drawback to applying our procedure in this respect. We improved the discussion to explain why the π -average of the overlaps may remain large even for nonreversible processes, and we incorporated a discussion on the diameter-based lower bounds of both classical and quantum algorithms to delineate processes that cannot be accelerated. We argue that such processes are not the most relevant in typical application scenarios. We also added further references to clarify our oracle model assumptions.

To summarize the changes made to address the reviewer's comments: we refined the scope of our claims by explicitly stating that our focus is on processes whose stationary distribution is known only up to a multiplicative constant; precisely the setting in which Markov Chain Monte Carlo methods are most widely employed. We now cite the Quantum Rejection Sampling (QRS) method as a relevant related approach. A detailed head-to-head comparison of generic methods, including ours, is generally only possible for specific instances and would merit a dedicated study. We have adjusted the text to make this limitation clear.

Referee 1

Comment: *I thank the authors for their efforts in revising the manuscript and addressing my concerns.*

Response: We thank the reviewer for their comments and for helping to improve the presentation of the manuscript.

Referee 4

Comment: *After reading the reply to referees, I must say that the paper seems now much clearer to me; and hopefully this is representative of more general readers' opinion. Most arguments are clarified and I appreciate the additions about the context.*

There is just one technical point which I found too sketchy, namely about the oracle access. Indeed, the reference provided explains how to use the oracle of a reversible Markov chain, while here the purpose is precisely to treat non-reversible ones. It would seem necessary to explain how, having access to the "P" oracle, one would construct the "P" oracle. Indeed, in general, it is not obvious (to me) that being able to predict where the walk will go next, implies you could as easily deduce where the walk is coming from.*

I may just be overlooking some obvious property; or maybe not. If this point was clarified, all my comments would be resolved.

Response: We thank the reviewer for their insightful and constructive feedback. We believe that addressing their comments has significantly improved the clarity of the manuscript.

Whenever the transition probabilities $P(x, y)$ are computable and we can compute $\alpha\pi(x)$ for some positive constant α , we argue that we can efficiently compute the transition probabilities of P^* and, as a consequence, access it through the same oracle model. Indeed,

$$P^*(x, y) = \frac{\pi(y)}{\pi(x)} P(y, x) = \frac{\alpha\pi(y)}{\alpha\pi(x)} P(y, x). \quad (1)$$

In the particular case of kinetic processes, which encompass many practical applications of nonreversible processes for sampling, it is often the case that $P^* = FPF$, where F is the operator that reverses the velocity component of the process ($v \rightarrow -v$).

The paragraph "*Encoding Markov chains in quantum computers*" now concludes with the following clarification: "*The operators R and R^* are typically constructed with arithmetic oracles capable of performing the transformation*

$$|x\rangle \rightarrow \sum_{y \in \mathbb{S}} \sqrt{P(x, y)} |x, y\rangle, \quad (2)$$

for every state $x \in \mathbb{S}$ (see, for example, [citation]). Such oracles are efficiently implementable whenever the transition probabilities are efficiently computable. If the stationary distribution π can be computed up to a multiplicative constant, ratios of the form $\pi(y)/\pi(x)$ can be efficiently evaluated for any pair of states $x, y \in \mathbb{S}$. Consequently, the time-reversed transition probabilities $P^(x, y) = P(y, x)\pi(y)/\pi(x)$ can also be computed efficiently, implying that P^* is accessible under the same oracle model. As demonstrated in [citations] and exemplified in the Appendix, it is often possible to construct these operators much more efficiently. For kinetic processes, where the state space includes both position and velocity components (as in underdamped Langevin dynamics), P^* can be efficiently implemented by reversing the initial velocity, applying one step of P , and then reversing the velocity again."*